# EphrinB2 drives perivascular invasion and proliferation of glioblastoma stem-like cells

Benjamin Krusche[1,2†], Cristina Ottone[1,2†], Melanie P Clements[1,2], Ewan R Johnstone[3,4], Katrin Goetsch[1,2], Huang Lieven[2,5], Silvia G Mota[6], Poonam Singh[7], Sanjay Khadayate[2], Azhaar Ashraf[1,2], Timothy Davies[1,2], Steven M Pollard[8], Vincenzo De Paola[2,5], Federico Roncaroli[7,9], Jorge Martinez-Torrecuadrada[6], Paul Bertone[3,4], Simona Parrinello[1,2*]

[1]Cell Interactions and Cancer Group, MRC Clinical Sciences Centre (CSC), London, United Kingdom; [2]Institute of Clinical Sciences (ICS), Faculty of Medicine, Imperial College London, London, United Kingdom; [3]Wellcome Trust - Medical Research Council Stem Cell Institute, University of Cambridge, Cambridge, United Kingdom; [4]European Molecular Biology Laboratory, European Bioinformatics Institute, Hinxton, United Kingdom; [5]Neuroplasticity and Diseases Group, MRC Clinical Sciences, London, United Kingdom; [6]Proteomics Unit, Centro Nacional de Investigaciones Oncologicas, Madrid, Spain; [7]Department of Histopathology, Imperial College Healthcare Trust, London, United Kingdom; [8]MRC Centre for Regenerative Medicine, The University of Edinburgh, Edinburgh, United Kingdom; [9]Wolfson Molecular Imaging Centre, University of Manchester, Manchester, United Kingdom

*For correspondence: simona. parrinello@imperial.ac.uk

†These authors contributed equally to this work

Competing interests: The authors declare that no competing interests exist.

**Abstract** Glioblastomas (GBM) are aggressive and therapy-resistant brain tumours, which contain a subpopulation of tumour-propagating glioblastoma stem-like cells (GSC) thought to drive progression and recurrence. Diffuse invasion of the brain parenchyma, including along preexisting blood vessels, is a leading cause of therapeutic resistance, but the mechanisms remain unclear. Here, we show that ephrin-B2 mediates GSC perivascular invasion. Intravital imaging, coupled with mechanistic studies in murine GBM models and patient-derived GSC, revealed that endothelial ephrin-B2 compartmentalises non-tumourigenic cells. In contrast, upregulation of the same ephrin-B2 ligand in GSC enabled perivascular migration through homotypic forward signalling. Surprisingly, ephrin-B2 reverse signalling also promoted tumourigenesis cell-autonomously, by mediating anchorage-independent cytokinesis via RhoA. In human GSC-derived orthotopic xenografts, *EFNB2* knock-down blocked tumour initiation and treatment of established tumours with ephrin-B2-blocking antibodies suppressed progression. Thus, our results indicate that targeting ephrin-B2 may be an effective strategy for the simultaneous inhibition of invasion and proliferation in GBM.

## Introduction

Glioblastoma (GBM) is the most common and aggressive type of primary brain tumour and one of the most lethal types of cancer (*Chen et al., 2012b*). Current therapies consist of maximally safe surgical resection followed by radio and chemotherapy. However, these are largely ineffective and

**eLife digest** Glioblastoma is the most common and deadly type of brain cancer. On average, patients with glioblastoma only survive 15 months even with aggressive treatment. One of the main reasons that therapy fails is the strong tendency of the tumor cells in this form of cancer to spread into the normal brain tissue. This makes it impossible for surgeons to completely remove the tumor, which means that the disease will almost always recur.

Within the brain, invading glioblastoma tumor cells spread along pre-existing structures, like blood vessels. When the tumors use blood vessels as a highway to the rest of the brain it is called "perivascular invasion". Scientists do not know exactly how glioblastoma cells move along the blood vessels. Learning more about this type of tumor cell movement could help scientists develop treatments to stop the tumor cells from spreading.

Now, Krusche et al. show that the glioblastoma tumor cells highjack the system that normally helps keep brain cells in place. Experiments with mouse and human tumor cells grown in the laboratory and injected in mice to produce glioblastoma tumors showed that a family of proteins called ephrins determines whether perivascular invasion occurs. Ephrins are found on the surface of both tumor cells and blood vessels. Normally, the blood vessels use these proteins to block the spread of normal brain cells. However, tumor cells override this normal anti-tumor mechanism and become able to spread along the blood vessels.

Specifically, Krusche et al. showed that an increase in the levels of ephrin-B2 in tumor cells allows them to move along the blood vessels. Ephrin-B2 was also found to drive the multiplication of the tumor cells independently of the protein's interactions with the blood vessels. Using antibodies to block ephrin-B2 in tumors greatly reduced tumor size and extended survival in mice with glioblastoma tumors. The experiments suggest the blocking ephrin-B2 might be a therapy that both stops the glioblastoma cells from spreading and prevents the tumor cells from multiplying.

GBM invariably recur following treatment, resulting in a median survival of <15 months (*Weathers and Gilbert, 2014*).

A leading cause of GBM recurrence is the diffuse infiltration of the tumour cells into the surrounding brain parenchyma, which limits efficacy of surgery and radiotherapy (*Cuddapah et al., 2014*; *Lamszus et al., 2003*). In addition, GBM contain subpopulations of cells with stem cell properties termed glioblastoma stem-like cells (GSC), which are able to self-renew, differentiate into tumour-bulk cells and reconstitute a phenocopy of the original lesion upon transplantation (*Venere et al., 2011*). It is therefore increasingly recognized that GSC are critical players in tumour initiation and progression. Furthermore, GSC were shown to be intrinsically resistant to chemo- and radiotherapy and more invasive than non-stem tumour cells (*Chen et al., 2012a*; *Cheng et al., 2011*; *Bao et al., 2006*; *Venere et al., 2011*; *Sadahiro et al., 2014*). This suggests that GSC might be the primary cells within GBM responsible for infiltration and tumour recurrence following therapy and that GSC-targeting therapies should be beneficial for GBM treatment.

GBM invasion occurs along three main routes: the white matter tracts, the interstitial space of the brain and the perivascular space surrounding blood vessels (*Scherer, 1938*; *Cuddapah et al., 2014*). Invasion along the perivascular space is a favourable migration route because endothelial cells secrete chemoattractants, which actively recruit tumour cells to the vasculature (*Montana and Sontheimer, 2011*; *Cuddapah et al., 2014*). In addition, the perivascular space is enriched in migration-promoting ECM components and is fluid-filled, thereby opposing little physical resistance to invading tumour cells (*Gritsenko et al., 2012*; *Cuddapah et al., 2014*). Importantly, within GBM, GSC are particularly prone to perivascular invasion, likely due to their similarities with normal neural progenitors, which preferentially migrate along blood vessels during development and after injury in the adult (*Cuddapah et al., 2014*; *Watkins et al., 2014*). In agreement with this, GSC reside in perivascular niches and the majority of invading cells migrate along the host vasculature in xenograft GSC models of both mouse and human origin (*Farin et al., 2006*; *Zagzag et al., 2008*; *Baker et al., 2014*; *Calabrese et al., 2007*; *Winkler et al., 2009*; *Watkins et al., 2014*).

Besides infiltration, GBM/vascular interactions underlie two additional key tumourigenic mechanisms. First, tumour cell migration along pre-existing normal blood vessels is an important tumour vascularisation mechanism, known as vascular co-option, by which tumours gain access to oxygen and nutrients independent of angiogenesis (*Donnem et al., 2013*). Vascular co-option plays crucial roles during initial tumour growth, seeding of satellite lesions and tumour recurrence following therapy. Second, association with the perivascular niche provides GSC with important self-renewal and survival signals, which support GSC tumour-propagating abilities and therapeutic resistance (*Charles and Holland, 2010*). Therefore, interactions with the vasculature, particularly within the GSC compartment, are critical throughout gliomagenesis and were indeed proposed as promising therapeutic targets for GBM treatment, but the mechanisms remain poorly defined (*Cuddapah et al., 2014*; *Vehlow and Cordes, 2013*).

GSC share many properties with normal neural stem cells (NSC), such as stem cell markers expression (Sox2, Nestin, CD133, ALDH1, etc.) and multilineage differentiation and, importantly, extensive evidence indicates that NSC themselves can be cells of origin in GBM (*Chen et al., 2012b*; *Venere et al., 2011*; *Chen et al., 2012a*). However, GSC also significantly differ from their normal counterparts, in that they harbour transforming mutations that drive their tumourigenic properties, including deregulated proliferation and increased invasiveness (*Engstrom et al., 2012*). Therefore, the comparison of GSC carrying known mutations to otherwise genetically matched normal NSC, should inform disease mechanisms, enable genotype-phenotype correlation and may identify new therapeutic targets for GBM.

In this study, we generated a murine GBM model by sequentially introducing oncogenic lesions relevant to the human disease into normal NSC and compared vascular interactions of the resultant transformed GSC-like cells and immortalised parental cells to interrogate mechanisms of perivascular invasion. Using this system, we identified ephrin-B2 as a critical driver of perivascular invasion. ephrin-B2 is a member of the Eph/ephrin family of receptor tyrosine kinases and their membrane-bound ligands, a fundamental cell communication system with widespread roles in tissue development, maintenance and disease (*Pasquale, 2010*). Activation of Eph receptors by ephrin ligands on adjacent cells modulates cell behaviour, including migration, proliferation and stemness, by eliciting forward signalling downstream of Ephs and reverse signalling downstream of ephrins (*Kullander and Klein, 2002*). Deregulation of the Eph/ephrin system contributes to the pathogenesis of many types of cancer, including GBM (*Pasquale, 2010*). Indeed, EphA2 and EphA3 were shown to drive GSC self-renewal, whereas EphA2, EphA4, EphB2, ephrin-B2 and ephrin-B3 have all been linked to GBM invasion, through incompletely understood mechanisms (*Day et al., 2014*; *Tu et al., 2012*; *Binda et al., 2012*; *Day et al., 2013*; *Nakada et al., 2010*; *2011*).

We found that ephrin-B2 expressed on vascular endothelial cells inhibits the migration of non-tumourigenic cells, resulting in cell confinement. In contrast, transformation to GSC-like cells overrides this tumour-suppressive mechanism to drive perivascular invasion. We show that this is caused by upregulation in GSC-like cells of the same ephrin-B2 ligand, which desensitizes the cells to vascular confinement by constitutively activating Eph forward signalling non-cell-autonomously. Furthermore, we discovered that ephrin-B2 reverse signaling also elicits tumour cell proliferation in the absence of normal anchorage signals by driving Rho-A-dependent cytokinesis in a cell-autonomous manner. Consistent with these important roles, ephrin-B2 overexpression was sufficient to fully transform immortalised NSC to the same extent as oncogenic Ras. In human GBM specimens, high Ephrin-B2 levels were detected in perivascular tumour cells with GSC features at the infiltrative tumour margin, indicative of a role in the GSC compartment in primary tumours. Remarkably, *EFNB2* knockdown in primary human GSC isolated from patient material or treatment of established tumours derived from these GSC with anti-ephrin-B2 single chain blocking antibodies strongly suppressed tumourigenesis, by concomitantly inhibiting vascular association and proliferation. Thus, ephrin-B2 may be an attractive therapeutic target for the treatment of GBM.

## Results

### Endothelial ephrin-B2 compartmentalises immortalized, but not transformed, neural stem cells

To investigate mechanisms of GSC/vascular interactions in the context of syngeneic, immuno-competent brains, we sequentially introduced mutations commonly found in human GBM (RTK activation,p53 and RB inactivation) in primary murine SVZ NSC to generate fully transformed, GSC-like cells and genetically-matched immortalised NSC (*Network, 2008*). We used two complementary strategies for this. First, we used a 'classical' transformation paradigm previously shown to drive gliomagenesis in vivo, whereby NSC were immortalised with SV40 large-T antigen (imNSC1) and transformed with RasV12 (herein referred to as GSC1) to inactivate *Trp53* and *Rb*, and mimic the increased Ras signalling that results from *Nf1* loss, respectively (*Blouw et al., 2003*; *Hahn et al., 1999*; *Sonoda et al., 2001*; *Huszthy et al., 2012*). This approach allowed us to readily test candidate effectors by transforming NSCs isolated from mice carrying the specific mutation, as previously reported (*Blouw et al., 2003*). In the second approach, we induced transformation by defined genetic changes in the same pathways to rule out artifacts of oncogene overexpression. *Nf1*<sup>fl/fl</sup> NSCs were immortalised with p53 shRNAs and ectopic CDK4 to inactivate p53 and the p16/RB axis, respectively (imNSC2), and transformed by Cre-mediated *Nf1* deletion (herein referred to as GSC2).

Unlike previously reported for SVZ NSC in vitro (*Wang et al., 2012*), increased Ras signalling did not cause premature glial differentiation of the NSC in vitro in either model (*Figure 1—figure supplement 1a* and *Tables 1* and *2* and *Supplementary files 1* and *2*). In contrast, GSC1 and GSC2 retained stem cell properties in vitro as judged by high clonal efficiency in neurosphere culture and differentiation into glial and neuronal lineages upon mitogen withdrawal (*Figure 1—figure supplement 1a–c*). Furthermore, both cell types (but not their immortalised controls) formed colonies in soft agar (*Figure 4c,d*) and gave rise to highly aggressive tumours upon intracranial transplantation in immunocompromised mice (5/5 animals, 100% penetrance, median survival 24d for GSC1 and 38.5 for GSC2), whereas GSC1 also did so in syngeneic animals (5/9 animals, 56% penetrance, median survival 73d), indicative of a more aggressive phenotype. Consistent with their stem-like properties, clonal dilution experiments revealed that as little as 100 cells of either line was sufficient to generate aggressive tumours (*Figure 1—figure supplement 1d*). Importantly, the tumours recapitulated the histopathology and gene expression signatures of human GBM, including presence of necrosis, neovascularization, nestin and Sox2 expression and a strong enrichment in the Verhaak mesenchymal subtype gene signature (*Figure 1a–d* and *Figure 1—figure supplement 1e*) (*Kleihues, 2000*; *Verhaak et al., 2010*). The tumours presented diffuse borders with the majority of invading cells migrating along blood vessels and displacing astrocyte endfeet and pericytes to come in direct contact with endothelial cells, as previously reported for both murine and human GSC (*Figure 1b* and *Figure 1—figure supplement 1f*) (*Zagzag et al., 2008*; *Farin et al., 2006*; *Nagano et al., 1993*; *Watkins et al., 2014*). Thus, GSC1 and 2 resemble mesenchymal glioblastoma stem-like cell lines and are highly similar, interchangeable model systems.

Next, as only GSC1 formed tumours in syngeneic mice, we used this model to assess interactions with the normal brain vasculature in real time in vivo using 2-photon microscopy. GFP-labeled GSC1 or imNSC1 were injected at a mimimum depth of 150–200 µm into the cortex of syngeneic recipients under a chronic cranial window and 5–7 days later their migration in relation to Dextran-TxRed-labeled blood vessels was imaged over a period of 6 hr (*Figure 1e,f* and *Figure 1—figure supplement 1g*) (*Holtmaat et al., 2009*). Although imNSC do not form tumors long-term, the cells were still present, viable and migratory at this time-point, as indicated by morphology and motility outside of the perivascular space. Consistent with previous reports, both cell types readily associated to the vasculature (*Kokovay et al., 2010*; *Baker et al., 2014*; *Farin et al., 2006*; *Nagano et al., 1993*; *Watkins et al., 2014*; *Winkler et al., 2009*). However, while individual GSC1 migrated out of the tumour mass along blood vessels, resulting in cell scattering, imNSC remained in stationary groups (*Figure 1e,f*). GSC1 perivascular migration was independent of their position relative to the tumour bulk or cell density, with single cells migrating from the main tumour as efficiently and at similar speed as GSC1 migrating at a distance from the tumour margin (*Figure 1e* and *Figure 1—figure supplement 1h*). Similarly, rare sparse imNSC further away from the tumour margin failed to migrate along blood vessels (*Figure 1—figure supplement 1h*). Importantly, this effect was not due to a general impaiment in imNSC migration, as these cells migrated to a similar extent as GSC in vitro

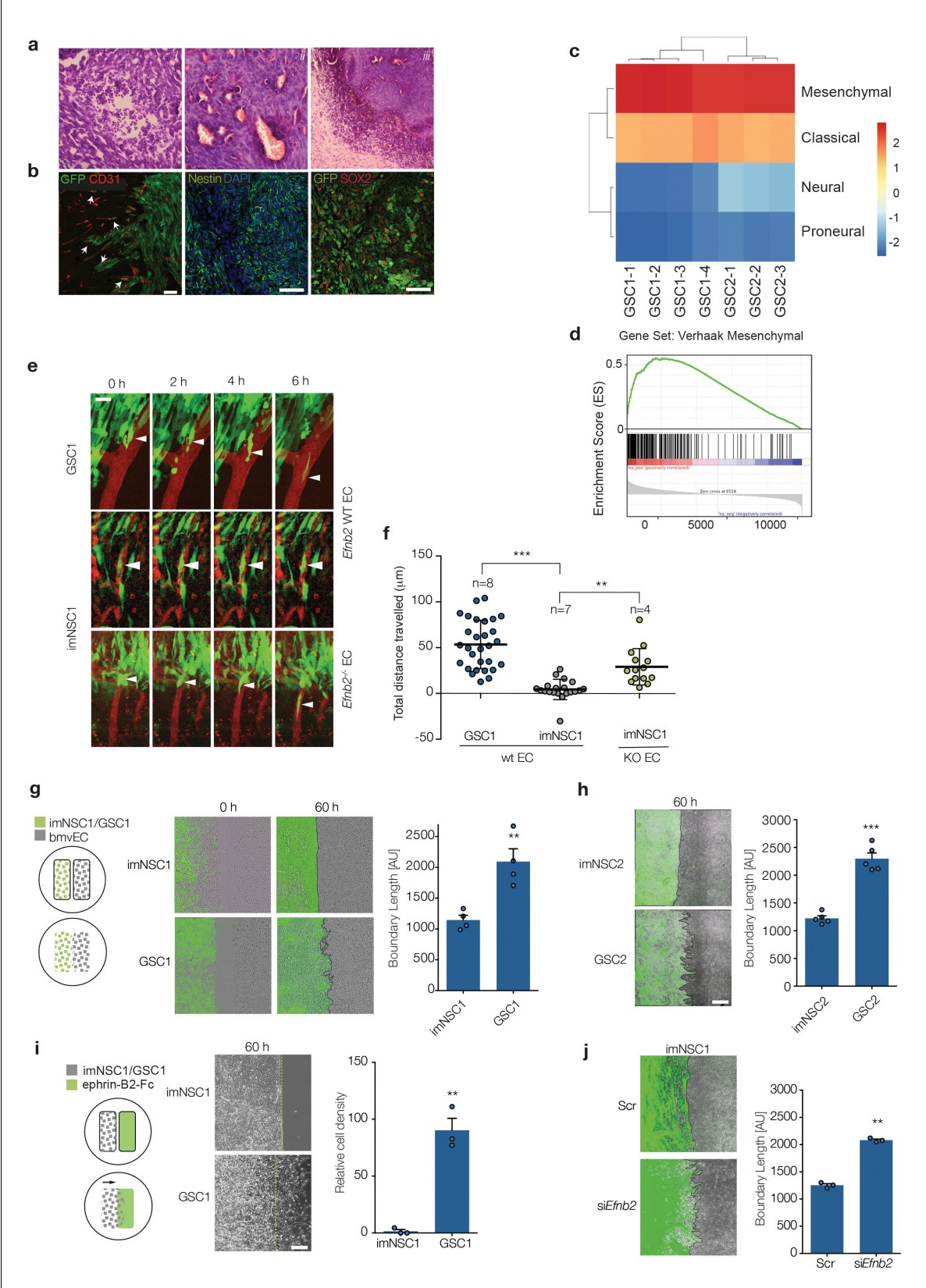

**Figure 1.** The vasculature compartmentalises immortalised neural but not glioma stem cells through endothelial ephrin-B2. (a) Representative H&E staining of GSC1 tumours 60 days after intracranial injection into nude mice. Neovascularisation (*i*), focal necrosis (*ii*) and increased cellular density (*iii*)

*Figure 1 continued on next page*

*Figure 1 continued*

can be observed. (**b**) Representative fluorescent images of GFP-labeled GSC1 tumours stained for the vascular marker CD31 (red) and GFP to identify tumour cells (green, left), the stem cell markers nestin (green, middle) and Sox2 (red, right), as indicated. The arrows indicate examples of GFP-labeled GSC1 that have migrated away from the tumour mass and are invading along the vasculature as single cells or in small groups. Scale bars = 50 μm. (**c**) Heatmap showing Verhaak subtype classification of 4 GSC1 and 3 GSC2-derived tumours in vivo. Colour scale with corresponding normalised enrichment scores is shown on the right. All tumours classified as mesenchymal with a nes >2.6. (**d**) Mean GSEA enrichment plot for the Verhaak mesenchymal gene signature in the GSC lines. Both nom p-val and FDR q-val = 0. (**e**) Intravital 2-photon micrographs of GFP-labeled imNSC1 and GSC1 cells injected into the cortex of wildtype (*Efnb2* WT EC) and endothelial specific *Efnb2* knockout mice (*Efnb2⁻/⁻* EC) and imaged 7 days later over 6 hr through an intracranial window at a depth of 200 μm. The vasculature was labeled by tail-vein injection of Tx-red conjugated Dextrans (3000 MW). Arrowheads indicate representative perivascular migration patterns for each genotype. Scale bar = 50 μm. (**f**) Quantification of the migrated distance of the cells depicted in (**c**). Each dot represents one cell. n indicates number of animals imaged. One way ANOVA with Tukey post hoc test. (**g**) Left: schematic representation of the experimental set up for in vitro migration assays with endothelial cells. Middle: merged fluorescent and phase contrast still images taken from time-lapse microscopy experiments of GFP-labeled imNSC1 and GSC1 (green) migrating towards brain microvascular endothelial cells (bmvEC, unlabeled cells) at the indicated time points. Right: quantification of boundary length at 60 hr. Students t-test. Scale bar = 500 μm. (**h**) Still fluorescence and phase contrast merged images of GFP-labeled imNSC2 and GSC2 migrating towards bmvEC (unlabeled, left) for 60 hr and quantification (right) as in (**e**). Error bars denote s.e.m., Students t-test. Scale bar = 500 μm. (**i**) Schematic representation of the experimental set up for in vitro migration assays toward recombinant ephrin-B2-Fc (left), phase contrast images (middle) and quantification (right) of imNSC1 and GSC1 migration against coated ephrinB2-Fc pre-clustered with fluorescently-labelled anti-Fc antibodies at 60 hr. Error bars denote s.e.m., Students t-test. Scale bar = 500 μm. Green dots denote boundary of ephrin-B2 coating identified by fluorescence. (**j**) Still images (left) and quantification (right) of GFP-labeled imNSC (GFP) migrating towards bmvEC (unlabeled) treated with control siRNA (Scr) or siRNA against *Efnb2* (si*Efnb2*) for 60 hr. Scale bar = 500 μm. Error bars denote s.e.m., Students t-test. For this and later figures dots indicate individual data points and ***p<0.001; **p<0.01 and *<0.05. See also *Figure 1—figure supplement 1* and *Figure 1—source data 1*.

The following source data and figure supplements are available for figure 1:

**Source data 1.** Raw data for all quantification of NSC/GSC migrated distance and boundary length shown in *Figure 1*.
**Figure supplement 1.** GSC1/2 resemble glioma stem-like cells.
**Figure supplement 1—source data 1.** Raw data for all quantitative analyses shown in *Figure 1—figure supplement 1*.

(*Figure 1—figure supplement 1i*). This suggests that signals from the vasculature compartmentalises non-tumourigenic cells to restrict their migration, whereas transformation overrides these signals to enable perivascular spread.

To test this more directly and identify potential effectors, we developed an in vitro cell migration assay that mimics the response of infiltrating GBM cells to initial contact with endothelial cells. We seeded both imNSC/GSC models and primary brain microvascular endothelial cells (bmvEC) in separate wells of culture inserts and assessed migration of the two cell types towards each other following insert removal by time-lapse microscopy. Remarkably, this assay closely recapitulated in vivo migratory patterns, in that endothelial cells strongly repelled and compartmentalised imNSC forming a sharp boundary (*Figure 1* and *Video 1*). Similar effects were observed with normal NSC, indicating that compartmentalisation is not caused by immortalisation, but rather reflects the response of normal cells to the vasculature (not shown). In contrast, GSC1 were refractory to compartmentalisation and migrated over the endothelial monolayer, giving rise to an uneven and longer boundary (*Figure 1g* and *Video 2*). The behavior of GSC2 was identical to GSC1, confirming that it is a

**Table 1.** Gene set enrichment analysis FDR q-values for astrocyte, oligodendrocyte, OPC and neuron signatures in GSC lines vs NSC.

|  | GAGE analysis q-value (FDR) | | | |
|---|---|---|---|---|
|  | GSCvsNS | GSC1vsNS | GSC2vsNS | Cell type term gene number |
| oligo | 1 | 1 | 1 | 497 |
| neuron | 1 | 1 | 1 | 497 |
| astro | 1 | 1 | 1 | 486 |
| opc | 1 | 1 | 1 | 490 |

**Table 2.** Gene set enrichment analysis FDR q-values for astrocyte, oligodendrocyte, OPC and neuron signatures in NSC vs GSC lines.

| | GAGE analysis q-value (FDR) | | | |
| --- | --- | --- | --- | --- |
| | NSvsGSC | NSvsGSC1 | NSvsGSC2 | Cell type term gene number |
| astro | 4.06848E-22 | 2.84603E-23 | 2.04565E-19 | 490 |
| opc | 1.57235E-16 | 7.01306E-17 | 3.24522E-14 | 486 |
| neuron | 1.51727E-06 | 4.55938E-07 | 4.58202E-05 | 497 |
| oligo | 0.064920265 | 0.06708047 | 0.222744258 | 497 |

general property of transformed cells (*Figure 1h*). Eph/ephrin signalling is one of most important mediators of cell-cell contact-dependent boundary formation and we reported that endothelial ephrin-B2, the most abundant ephrin-B2 in the endothelial cells in vivo and in vitro (*Figure 1—figure supplement 1j*), induces cell sorting of normal NSC (*Cayuso et al., 2014*; *Ottone et al., 2014*; *Gale et al., 2001*). We therefore tested the role of ephrin-B2 in endothelial-induced compartmentalisation by assessing imNSC1 migration towards recombinant ephrin-B2-Fc. As shown in *Figure 1i* and *Videos 3* and *4*, ephrin-B2-Fc compartmentalised imNSCs, but not GSC, to the same extent as endothelial cells. This effect was highly specific and was not due to general differences in migration between imNSC and GSC cells because imNSC migrated onto Fc peptides, ephrin-A1 or ephrin-A5-Fc ligands to a similar extent as GSC cells (*Figure 1—figure supplement 1i*). Conversely, *Efnb2* knock-down in endothelial cells disrupted boundary formation against imNSC, indicating that endothelial ephrin-B2 is both necessary and sufficient for compartmentalisation (*Figure 1j* and *Figure 1—figure supplement 1j*).

To assess the role of vascular ephrin-B2 in vivo , we performed intravital imaging of imNSC implanted in inducible endothelial-specific conditional *Efnb2* knock-out mice (*Efnb2* [iΔEC]), following postnatal tamoxifen-mediated recombination (*Ottone et al., 2014*). Strikingly, selective deletion of ephrin-B2 in the endothelium elicited robust perivascular migration of imNSC1, confirming that vascular ephrin-B2 compartmentalises non-tumourigenic cells in vivo (*Figure 1e,f*).

## Upregulation of ephrinB2 drives perivascular invasion of GSC

As we reasoned that changes in Eph/ephrin levels might underlie the ability of GSC to escape ephrin-B2-mediated vascular compartmentalisation, we interrogated expression levels of all Ephs and ephrins in imNSC1/2 and GSC1/2 by qRT-PCR (*Figure 2a* and *Figure 2—figure supplement 1a*). Normal NSC controls were included to rule out p53 and Rb-dependent changes unrelated to perivascular migration. We found that p53 and Rb inactivation did not change Eph/ephrin expression

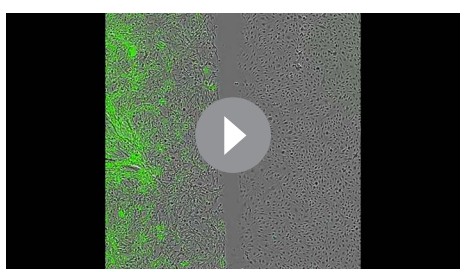

**Video 1.** Endothelial cells compartmentalise imNSC. Merged fluorescent and phase contrast movie from time-lapse microscopy experiments of cell-tracker labelled imNSC1 (green) migrating towards brain microvascular endothelial cells (bmvEC, unlabelled cells). Images were taken every 10 min for 60 hr.

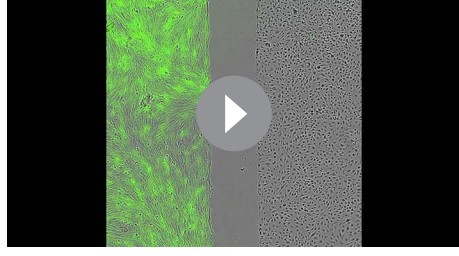

**Video 2.** GSC escape endothelial compartmentalisation. Merged fluorescent and phase contrast movie from time-lapse microscopy experiments of cell-tracker labelled GSC1 (green) migrating towards brain microvascular endothelial cells (bmvEC, unlabelled cells). A slight decrease in fluorescence was observed due to higher proliferation rate of the cells. Images were taken every 10 min for 60 hr.

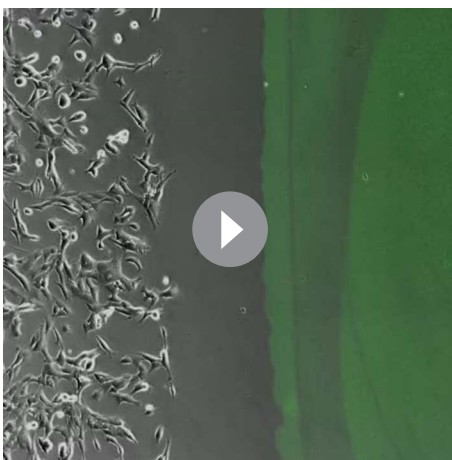

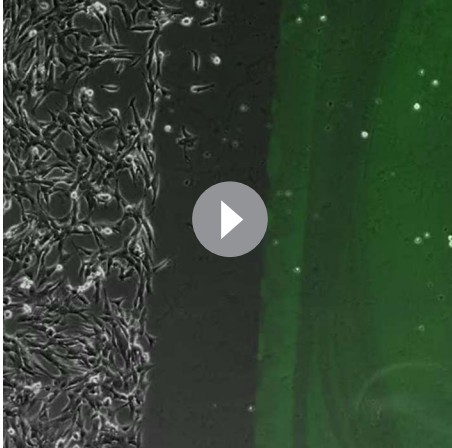

**Video 3.** imNSC are strongly compartmentalised by ephrinB2 ligand. Merged fluorescent and phase contrast movie from time-lapse microscopy of imNSC1 migrating towards coated ephrinB2-Fc labelled with fluorescent antibody (green). Images were taken every 10 min for 60 hr.

**Video 4.** GSC become insensitive to ephrinB2 ligand. Merged fluorescent and phase contrast movie from time-lapse microscopy of GSC1 migrating towards coated ephrinB2-Fc labelled with fluorescent antibody (green). Images were taken every 10 min for 60 hr.

substantially. In contrast, elevated Ras signalling, triggered by either RasV12 expression or *Nf1* loss, strongly downregulated three Ephs (*Epha4, Ephb1* and *Ephb2*) and upregulated two ephrins (*Efna5* and *Efnb2*) in both models. However, western analysis indicated that of these genes, only *Ephb1*, *Ephb2* and *Efnb2* changed significantly at the protein level (*Figure 2b*) and were thus further assessed in migration assays against ephrin-B2-Fc. Surprisingly, we found that re-introduction of *Ephb1* or *Ephb2* by overexpression in GSC1 (*Figure 2—figure supplement 1b*) did not affect migration towards recombinant ephrin-B2-Fc, indicating that changes in the complement of Eph receptors do not underlie unimpeded migration, as in other systems (*Figure 2c*) (*Astin et al., 2010*). Instead, genetic deletion of *Efnb2* in GSC1 (*Figure 2—figure supplement 1c*) fully rescued boundary formation in response to ephrin-B2-Fc and endothelial cells in vitro and blocked perivascular migration in vivo (*Figure 2c–e*). Conversely, *Efnb2* overexpression (*Figure 2—figure supplement 1d*) conferred imNSC1 with the ability to migrate over recombinant ephrin-B2 and a cultured endothelium and to escape vascular compartmentalisation in vivo (*Figure 2c–e*). Together, these results show that ephrin-B2 upregulation downstream of Ras underlies GSC evasion of ephrin-B2-mediated endothelial repulsion and invasion along the vasculature.

## Elevated ephrin-B2 desensitizes GSC to vascular repulsion by constitutively activating homotypic forward signalling

To understand the mechanisms involved in ephrin-B2-driven perivascular invasion, we characterised the early events of imNSC1 and GSC1 migration towards ephrin-B2-Fc in greater detail. We noticed that, although all GSC cells eventually migrated over ephrin-B2-Fc, a proportion of the cells (~20%) are repelled on first contact (*Figure 3a*). Repelled cells invariably migrated towards ephrin-B2-Fc either as single cells or in small groups (<2 contacts), whereas non-repelled cells migrated in larger groups (>3 contacts) (*Figure 3b*). This was not the case in imNSC cultures, where all cells were equally repelled, regardless of the number of homotypic cell-cell interactions at initial contact. This suggested that homotypic cell-cell interactions among GSC, mediated by intrinsic ephrin-B2, might underpin evasion of extrinsic ephrin-B2 repulsion. To test this, we repeated the migration assay under conditions that disrupt GSC cell-cell contacts by inhibiting cadherin-based junctions (*Ottone et al., 2014*). Both culture in low Ca$^{2+}$ media and dominant-negative N-cadherin overexpression completely disrupted cell-cell junctions without affecting cell motility (*Figure 3—figure supplement 1a,b*) and blocked GSC migration over ephrin-B2 ligands, indicating that homotypic cell-cell interactions are specifically required (*Figure 3c*).

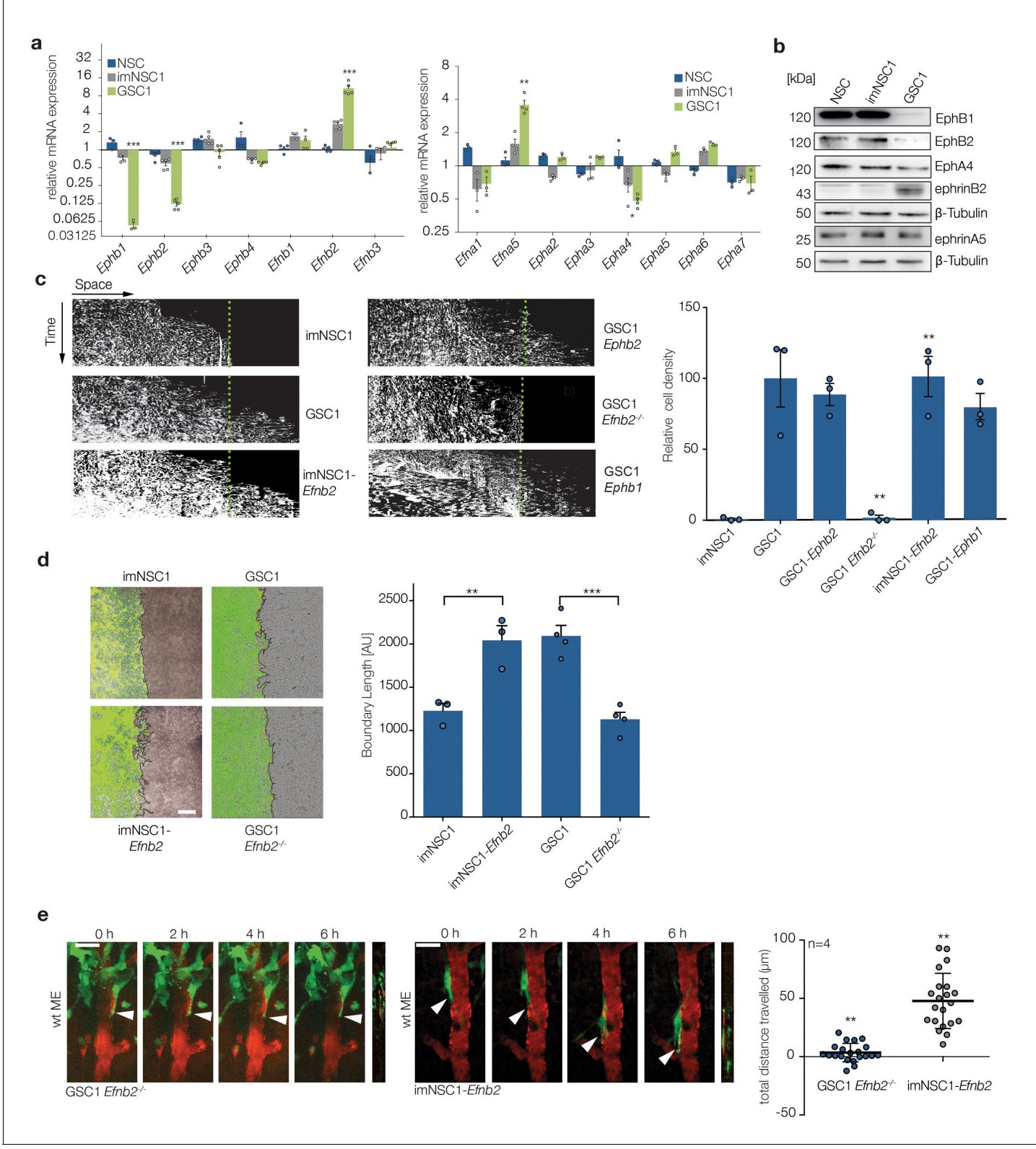

**Figure 2.** Upregulation of ephrin-B2 in GSCs enables perivascular invasion (**a**) Quantitative RT-PCR analysis of indicated Eph receptors and ephrin ligands. Error bars denote s.e.m., p values of differences in expression between imNSC1 and GSC1 are shown. Multiple t-test analysis. (**b**) Western analysis of levels of the indicated proteins in normal neural stem cells (NSC), imNSC1 and GSC1. n = 3 (**c**) Kymographs from time-lapse experiments of the indicated cell types migrating against coated ephrinB2-Fc over 60 hr. Quantification of the kymographs is shown on the right. Error bars denote s.e.

*Figure 2 continued on next page*

*Figure 2 continued*

m., one way ANOVA with Tukey post hoc test. (**d**) Overlaid fluorescent and phase contrast images (left) and quantifications (right) of boundaries formed at 60 hr of GFP-labeled imNSC1/GSC1 (green) migrating towards bmvEC (unlabeled) as indicated. Error bars denote s.e.m., one way ANOVA with Tukey post hoc test. Scale bar = 500 μm. (**e**) Representative 2-photon microscopy micrographs of GFP-labeled imNSC1-*Efnb2* and GSC1*Efnb2^-/-* imaged as in 1e. Quantification of migrated distances is shown on the right. Dots represent single cells measured across multiple animals. One way ANOVA with Tukey correction. p values are given compared to GSC1 for GSC1*Efnb2^-/-* and imNSC1 for imNSC1-*Efnb2*. Scale bar = 50 μm. See also *Figure 2—figure supplement 1* and *Figure 2—source data 1*.

The following source data and figure supplements are available for figure 2:

**Source data 1.** Raw data for qRT-PCR analysis and quantifications of kymographs, boundary assays and migrated distance in vivo of NSC/GSC cells shown in *Figure 2*.

**Figure supplement 1.** GSC2 show similar changes in Eph/ephrin levels as GSC1.

**Figure supplement 1—source data 1.** Raw data for all quantitative analyses shown in *Figure 2—figure supplement 1*.

We next asked whether homotypic ephrin-B2 signalling drives migration through activation of forward or reverse signalling within the GSC population. Ectopic expression of a C-terminal truncated form of ephrin-B2 (ΔC*Efnb2*), which lacks reverse signalling (*Pfaff et al., 2008*), promoted imNSC migration over ephrin-B2-Fc to the same extent as full-length *Efnb2*, demonstrating that forward signalling between neighboring cells is responsible (*Figure 3d* and *Figure 2—figure supplement 1d*). Indeed, western analysis (*Figure 3e*) indicated that while imNSC controls displayed low basal levels of activated Eph receptors (p-Eph), GSC exhinited constitutively high levels of p-Eph, despite their reduced levels of EphB1 and EphB2 (*Figure 2b*) and no increase in other ephrin-B2 receptors in these cells (*Figure 2a* and not shown). High basal p-Eph could be mimicked by ephrin-B2 overexpression in imNSC and reduced to basal control levels by culture in low $Ca^{2+}$ (*Figure 3e*). In addition, the high phosphorylation levels of Ephs in GSC appeared to be saturating, as, in contrast to imNSC, co-culture with endothelial cells or treatment with recombinant ephrin-B2 failed to further activate Eph receptors in these cells (*Figure 3f*). Importantly, Eph desensitization was again entirely dependent on endogenous ephrin-B2 because genetic deletion of *Efnb2* in GSC fully restored Eph stimulation by exogenous ephrin-B2. Consistent with the known ability of Ephs to trigger repulsion (*Astin et al., 2010*; *Pasquale, 2010*), such high basal levels of activated Eph resulted in GSC undergoing much greater scattering and faster migration than imNSC following homotypic collisions in sparse monocultures, in an ephrin-B2-dependent manner (*Figure 3—figure supplement 1c,d*). Thus, high ephrin-B2 levels in GSC result in constitutive activation of Eph forward signalling and desensitization to exogenous stimulation.

To further test whether this desensitization mechanism underlies the ability of GSC to override ephrin-B2-mediated repulsion, we pre-stimulated imNSC with recombinant ephrin-B2-Fc to stronly activate Eph forward signalling prior to their migration out of the insert (*Figure 3f*) and assessed migration towards coated ephrin-B2 in boundary assays. Parallel Fc-treated cultures served as controls. Remarkably, we found that ephrin-B2-Fc (but not Fc) pretreatment enabled migration over the ephrin-B2 boundary, confirming that activation of Eph forward signaling on first contact with exogenous ephrin-B2 is the mechanism involved. In agreement with these findings, immunofluorescence analysis of GSC1-derived orthotopic tumours revealed high pEph levels in cells migrating along blood vessels at the tumour edge, indicating that constitutive forward signaling also contributes to perivascular invasion in vivo (*Figure 3—figure supplement 1e*).

We conclude that GSC escape from endothelial compartmentalisation depends on continuous activation of Eph forward signalling elicited by elevated ephrin-B2 through homotypic cell-cell interactions within the tumour cell population. This, in turn, desensitises the receptors to further activation by extrinsic ephrins, thereby overriding the repulsion by endothelial ephrin-B2 and enabling unimpeded perivascular migration.

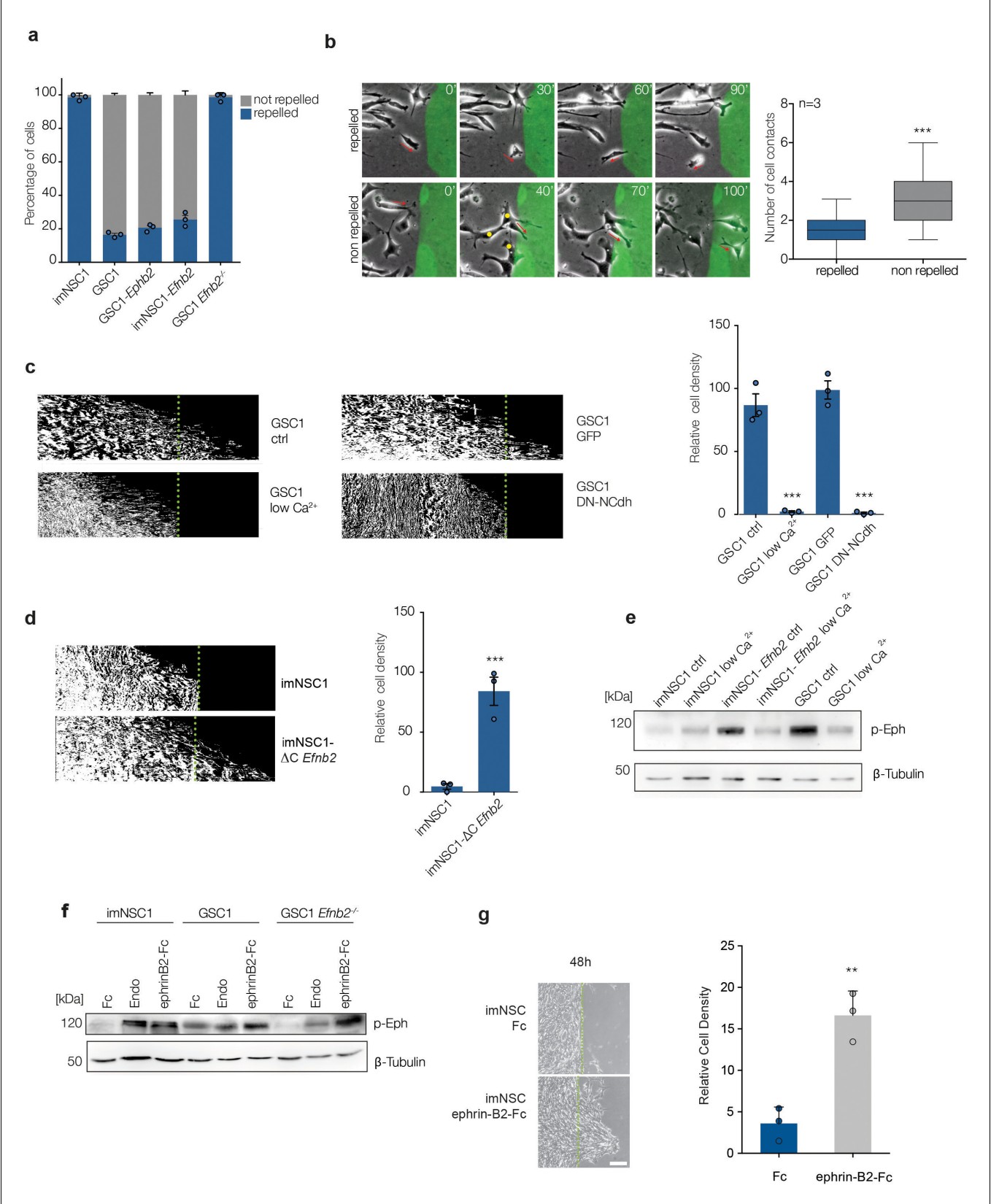

**Figure 3.** ephrin-B2 upregulation drives perivascular migration by saturating Eph forward signalling through homotypic cell-cell interactions. (a) Quantification of cell behavior upon initial contact with coated ephrin-B2. n = 3, error bars denote s.e.m. (b) Left: representative images taken from *Figure 3 continued on next page*

*Figure 3 continued*

videos of GSC1 cells as they first come in contact with coated ephrin-B2 (green). Right: quantification of the number of homotypic GSC1 cell-cell contacts at time of initial interaction with ephrin-B2. A minimum of 50 cells were counted in each experiment. Error bar denotes St.D, Student t-test. (c) Representative kymographs (left) and quantifications (right) of GSC1 migrating towards ephrin-B2 that were either cultured in normal media (ctrl) or low Ca$^{2+}$ conditions (low Ca$^{2+}$), or transduced with GFP control (GFP) or dominant negative N-Cadherin (DN-NCdh) adenoviral constructs. Dotted lines demarcate coated ephrin-B2 boundary. n = 3, error bars denote s.e.m., Student t-test. (d) Kymographs and quantifications of the responses of imNSC1 and imNSC1-ΔCEfnb2 to coated ephrin-B2 ligands. n = 3, error bars denote s.e.m., Student t-test. (e) Western analysis of the levels of activated Eph receptors (p-Eph) in the indicated cell types cultured in either normal growth media (ctrl) or calcium depleted (low Ca$^{2+}$) conditions. β-tubulin served as loading control. n = 3 (f) Western blots of p-Eph levels in indicated cells cultured either with control proteins (Fc), endothelial cells (Endo) or ephrinB2-Fc (ephrinB2-Fc) for 18 hr. β-tubulin is used as loading control. n = 3 (g) Phase contrast images (left) and quantification (right) of imNSC1 migration towards coated ephrin-B2-Fc following pre-treatment with either control (Fc) or clustered ephrinB2-Fc to activate Eph forward signalling. Experiments were stopped at 48 hr to assure maximal stimulation of the cells throughout the assay. Green dots denote boundary of ephrin-B2 coating identified by fluorescence. n = 3, error bars denote s.e.m., Students t-test. Scale bar = 250 μm. See also *Figure 3—figure supplement 1* and and *Figure 3—source data 1*.

The following source data and figure supplements are available for figure 3:

**Source data 1.** Raw data for all quantifications of NSC/GSC migration assays shown in *Figure 3*.
**Figure supplement 1.** Increased repulsion between ephrin-B2 expressing cells leads to greater migration velocity and distance.
**Figure supplement 1—source data 1.** Raw data for all quantitative analyses shown in *Figure 3—figure supplement 1*.

## ephrin-B2 transforms imNSC

Given these important roles in invasion, we asked whether ephrin-B2 might also affect tumourigenesis by performing tumourigenicity studies of luciferase-tagged imNSC1, imNSC1-*Efnb2*, GSC1 and GSC1*Efnb2*$^{-/-}$ implanted orthotopically in nude mice. We used immunocompromised mice for these experiments because the incomplete tumour penetrance of GSC in syngeneic animals precludes rigorous assessment of their tumourigenicity in the syngeneic model. Strikingly, quantitative imaging and survival analysis revealed that, while imNSC1 did not form tumours and GSC1 formed tumours at full penetrance, *Efnb2* deletion strongly suppressed GSC1 tumour growth and, conversely, *Efnb2* overexpression was sufficient to fully transform imNSC (*Figure 4a,b*). Specifically, imNSC1-*Efnb2* tumours resulted in a similar median survival as GSC1 (28d). In contrast, all imNSC animals survived tumour-free beyond 200 days and 7 out of 9 animals of the GSC1*Efnb2*$^{-/-}$cohort developed lesions with much slower kinetics, while the remaining 2 mice remained tumour-free, as confirmed by post-mortem examination. These effects were not due to a general loss of stem-like properties due to ephrin-B2 deletion, as clonal efficiency remained unaffected in GSC*Efnb2*$^{-/-}$ (*Figure 1—figure supplement 1b*). Thus, ephrin-B2 can substitute oncogenic Ras for transformation.

To dissect the mechanisms involved, we performed soft-agar assays, which assess proliferation in the absence of anchorage signals, a property closely linked to in vivo tumourigenicity (*Freedman and Shin, 1974*). Consistent with our results in vivo , the majority of imNSC1 and GSC1 *Efnb2*$^{-/-}$ remained as single cells in soft agar and only less than 5% of the cells generated small colonies, as previously reported for NSC, again indicating that immortalised cells behave like NSC (*Figure 4c*) (*Gursel et al., 2011*). Instead, imNSC1-*Efnb2* and GSC1 formed large colonies at similar high efficiency, indicating that ephrin-B2 drives anchorage-independent proliferation. Intriguingly, and in contrast to its role in perivascular invasion, ephrin-B2 effects on proliferation were dependent on reverse signalling and independent of homotypic cell-cell contact. Indeed, imNSC1-ΔC*Efnb2* did not form colonies in suspension and imNSC1-*Efnb2* overexpressing DN-Ncadherin formed colonies at similar efficiency as GFP-controls, demonstrating that Eph forward signalling is dispensable (*Figure 4c* and *Figure 4—figure supplement 1a*). To rule out the possibility that the transforming ability of ephrin-B2 might be a peculiar feature of the lgT-Ras model, we repeated the soft agar assays with imNSC2, imNSC2-*Efnb2*, imNSC2-ΔC*Efnb2*, GSC2 and GSC2 knock-down for *Efnb2* and obtained identical results, confirming the generality of these findings and the functional equivalence of the two GSC lines (*Figure 4d* and *Figure 4—figure supplement 1b*).

We next asked how ephrin-B2 overrides anchorage checkpoints by comparing cell cycle progression of immortalised and transformed cells in suspension. We seeded imNSC1, imNSC1-*Efnb2*,

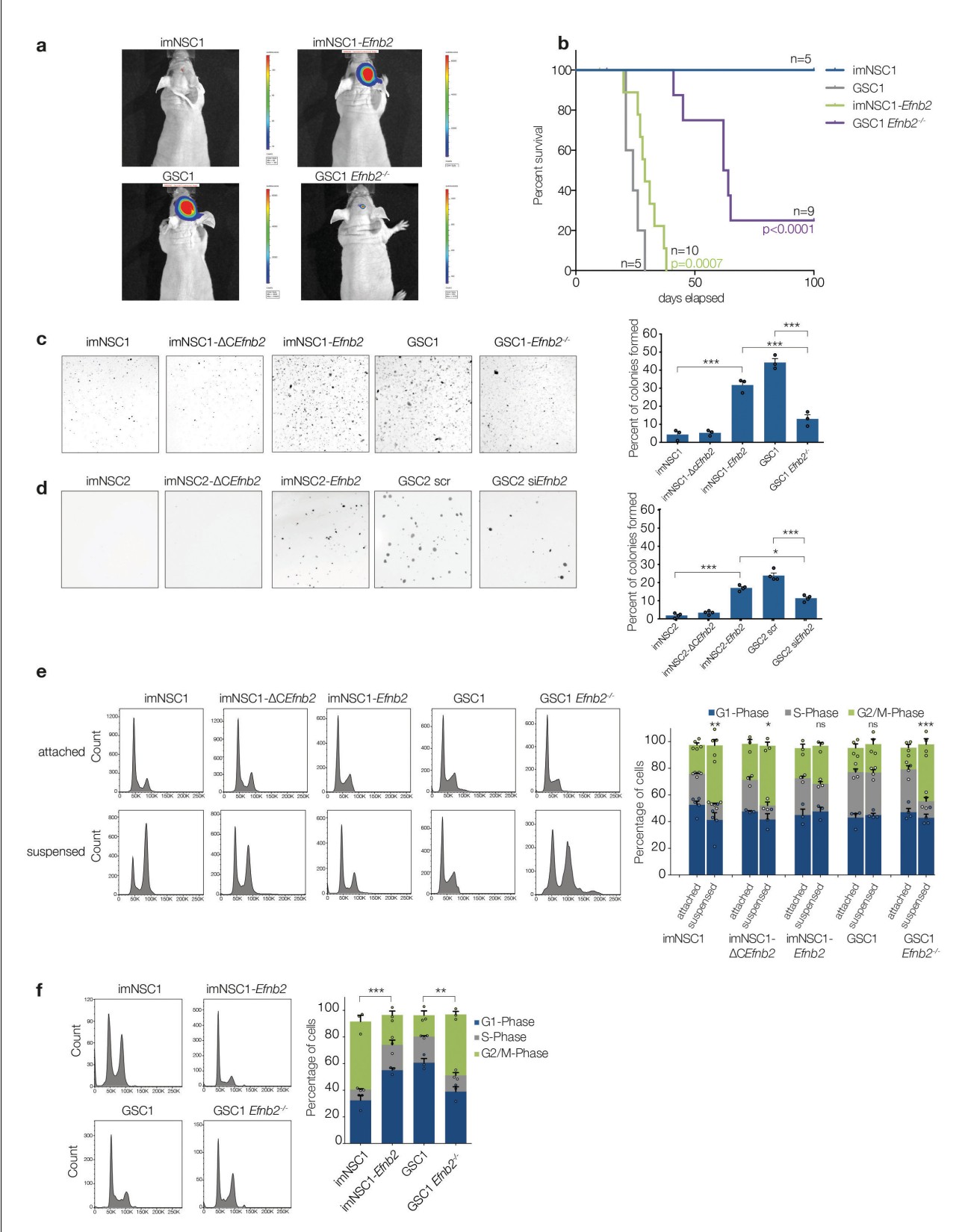

**Figure 4.** ephrin-B2 is a glioblastoma oncogene controlling anchorage independent proliferation. (**a**) Representative bioluminescent images of nude mice 20 days after intracranial injection of imNSC1, imNSC1-*Efnb2*, GSC1, GSC1*Efnb2*[-/-]. (**b**) Kaplan-Meier survival plots of the mice depicted in (**a**).
*Figure 4 continued on next page*

*Figure 4 continued*

Significance is given relative to imNSC1 for imNSC1-*Efnb2* and to GSC1 for GSC1-*Efnb2*$^{-/-}$. Log Rank Mantel Cox. (c, d) Left: Representative micrographs of GSC1 (c) and GSC2 (d) cells of indicated genotype grown in soft agar for 10d. Right: quantification of number of colonies formed in soft agar in all cultures, expressed as percentage over total number of seeded cells. n = 3, error bars depict s.e.m. One way ANOVA with Tukey correction. (e) Left: representative FACS profiles of cells grown in attachment or methylcellulose for 72 hr, showing DNA content by propidium iodide (PI) staining. Right: quantification of cell cycle phases from the FACS profiles. n = 3–5 as indicated by the dots. Error bars depict s.e.m., p values indicate significance of changes in G2/M phase. (f) Representative PI FACS profiles and quantifications of imNSC and GSC1 isolated from brain tissue 7 days after intracranial injection. n = 3. Error bars depict s.e.m. One way ANOVA with Tukey post hoc test shown for G2/M phase. See also *Figure 4— figure supplement 1* and *Figure 4— source data 1*.

The following source data and figure supplements are available for figure 4:

**Source data 1.** Raw data for Kaplan Meier analysis, number of colonies formed in soft agar and cell-cycle analysis presented in *Figure 4*
**Figure supplement 1.** Anchorage-independent proliferation is independent of homotypic cell-cell contacts.
**Figure supplement 1—source data 1.** Raw data for all quantitative analyses shown in *Figure 4—figure supplement 1*

GSC1 and GSC1 *Efnb2*$^{-/-}$ in adhesion or methylcellulose culture for 72 hr and measured their PI-FACS profiles (*Cremona and Lloyd, 2009*). As anticipated, all cells proliferated efficiently in attachment (*Figure 4e*). By contrast, their behavior in suspension was very distinct. While GSC1 continued to proliferate in methylcellulose with similar kinetics, suspended imNSC did not. This was not due to cell death as judged by activated caspase3$^+$ staining (*Figure 4—figure supplement 1c*). Instead, imNSC arrested with a 4n DNA content. This indicated that while imNSC progress normally through the G1 and S phases of the cell-cycle, their progression through G2/M is blocked by an anchorage checkpoint, which is overridden by activated Ras.

Importantly, Ras-mediated progression through G2/M was again dependent on ephrin-B2 reverse signalling, because *Efnb2* deletion in GSC re-instated the G2/M arrest (without additional apoptosis, *Figure 4—figure supplement 1c*), and the cell-cycle arrest of imNSC could be rescued by overexpression of full-length, but not ΔC, *Efnb2* (*Figure 4e*).

To assess the relevance of these findings to in vivo tumourigenesis, 7d after intracranial implantation imNSC1, imNSC1-*Efnb2*, GSC1 and GSC1*Efnb2*$^{-/-}$, were recovered from the injected brains and their PI profiles analysed (*Figure 4f*). Strikingly, the cell-cycle profiles of all cells were indistinguishable from the corresponding methylcellulose cultures, confirming that ephrin-B2 drives gliomagenesis in vivo by promoting proliferation in the absence of normal anchorage signals.

## ephrin-B2 drives anchorage-independent cytokinesis

A previous study reported that human fibroblasts cultured in suspension arrest at cytokinesis and that oncogenic Ras can bypass this arrest (*Thullberg et al., 2007*). We therefore explored the role of cytokinesis in our system by pulsing all attached and suspended cultures with EdU to distinguish G2/M arrested from cycling cells completing mitosis and staining for phalloidin to detect cortical actin. As shown in *Figure 5a and b*, we found that all methylcellulose cultures devoid of ephrin-B2 reverse signalling (imNSC1, imNSC1-ΔC*Efnb2* and GSC1 *Efnb2*$^{-/-}$) contained a much larger proportion of EdU$^-$ binucleated cells with decondensed chromatin compared to adherent conditions, indicative of a cytokinesis block (*Thullberg et al., 2007*). In contrast, cells with intact ephrin-B2 reverse signalling (imNSC1-*Efnb2* and GSC1) had similar low percentages of EdU$^+$ binucleated cells in both suspension and attachment culture. Thus, ephrin-B2 reverse signalling drives anchorage-independent cytokinesis.

Of the known effectors of ephrin-B2, Src and RhoA have been previously linked to anchorage independent proliferation, with RhoA also regulating constriction of the contractile ring during cytokinesis (*Daar, 2012*; *Desgrosellier et al., 2009*; *Jordan and Canman, 2012*). We therefore assessed Src and RhoA in ephrin-B2-mediated anchorage-independent cytokinesis by measuring levels of their activated forms (phosphorylated Src, p-Src, and GTP-bound RhoA) in suspension by western blot and pull-down assays, respectively. p-Src was undetectable in suspended imNSC1, imNSC-*Efnb2* and GSC1 and, similarly, levels of activated FAK, a critical Src substrate, were also reduced (*Figure 5—figure supplement 1a*), indicating that Src does not play a role in our system. In stark

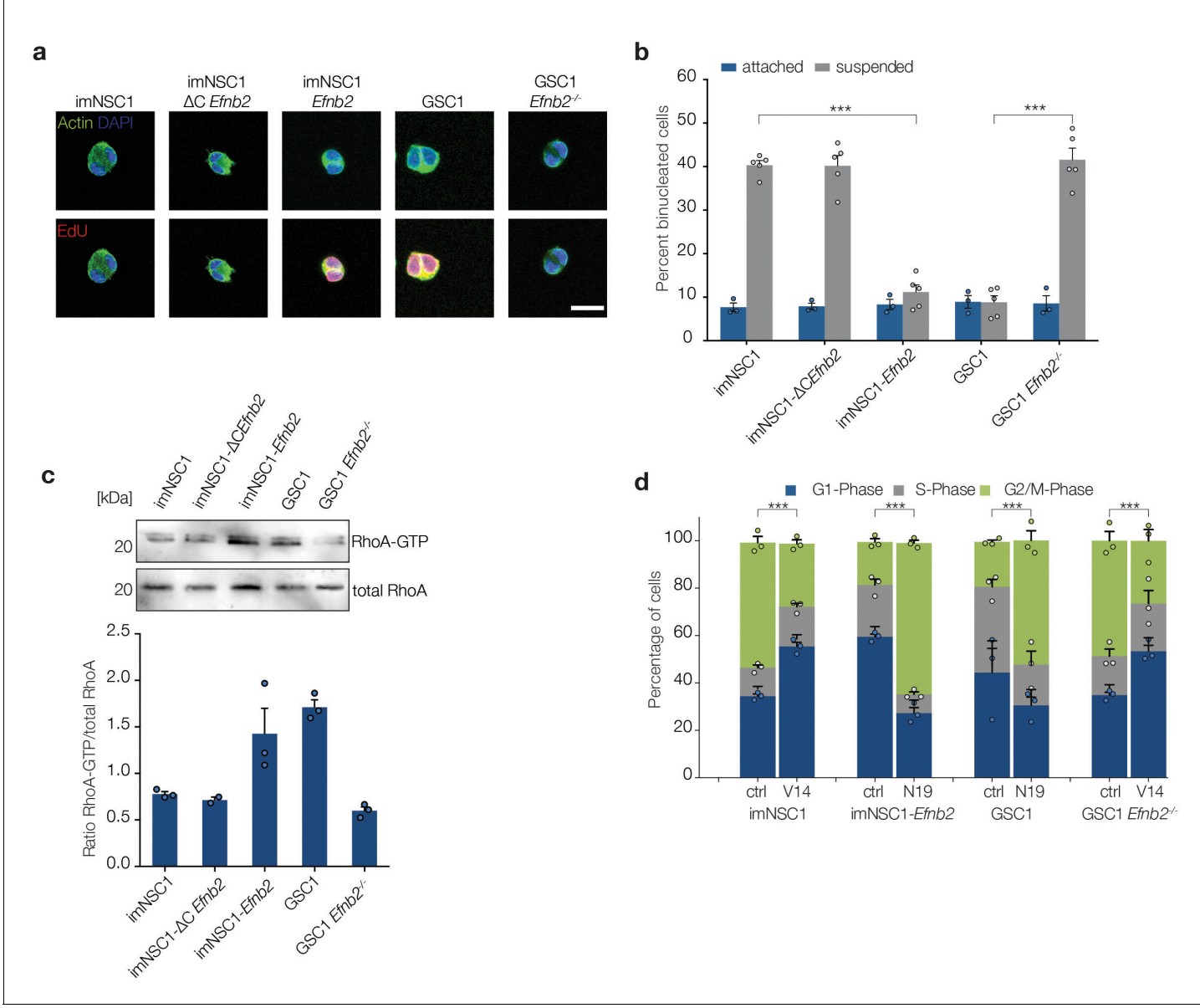

**Figure 5.** ephrin-B2 mediates cytokinesis in the absence of anchorage through RhoA. (a) Representative fluorescent images of binucleated cells grown in suspension for 24 hr and stained with phalloidin (green) to label cortical actin and EdU (red) to distinguish cycling from arrested cells. Scale bar = 20 µm. (b) Quantification of binucleated cells in indicated attached and suspended cultures. Values represent percentages over total number of cells. n = 4, error bars denote s.e.m. Two way ANOVA with Tukey post hoc test. (c) Western analysis of RhoA pulldown assays showing levels of activated RhoA (RhoA-GTP) and total RhoA levels in indicated cells grown in suspension for 24 hr . Bottom graph shows quantifications of activated RhoA levels from the western blots. n = 3. Error bars indicate s.e.m. (d) Quantification of the PI cell cycle profile as measured by FACS. Cells transfected with either constitutively active RhoA (V14) or dominant negative RhoA (N19) were grown in suspension for 24 hr . n = 3, error bars depict s.e.m. One way ANOVA with Tukey post hoc test was used to calculate p values for differences in G2/M phase of each suspended culture relative to corresponding attached cultures. See also *Figure 5—figure supplement 1* and *Figure 5—source data 1*.

The following source data and figure supplement are available for figure 5:

**Source data 1.** Raw data for quantifications of binucleated cells and cell cycle analysis presented in *Figure 5*.

**Figure supplement 1.** Src and FAK do not mediate anchorage-independent proliferation downstream of ephrin-B2.

contrast, levels of active RhoA were low in suspended cells that lacked ephrin-B2 reverse signalling (i.e. imNSC, imNSC1-ΔCEfnb2 and GSC1Efnb$^{-/-}$) but greatly increased in cells that express high levels of full-length ephrin-B2 (i.e. imNSC1-Efnb2 and GSC1), suggesting the involvement of RhoA (*Figure 5c*). To further test this, we overexpressed constitutively active (RhoA-V14, which increases levels of active GTP-bound RhoA, *Figure 5—figure supplement 1b*) and dominant negative (RhoA-N19, which inhibits activation of endogenous RhoA, *Figure 5—figure supplement 1b*) (*Qiu et al., 1995*) forms of RhoA in imNSC1/GSC1Efnb2$^{-/-}$ and imNSC1-Efnb2/GSC1, respectively, and assessed the ability of the cells to proliferate anchorage-independently by FACS. Remarkably, RhoA-V14 could rescue the cell-cycle arrest of imNSC1 and GSC1Efnb2$^{-/-}$, whereas RhoA-N19 arrested imNSC1-Efnb2 and GSC1 in G2/M (*Figure 5d* and *Figure 5—figure supplement 1c*). Together, these results demonstrate that ephrin-B2 drives anchorage-independent cytokinesis of GSCs through RhoA-mediated reverse signalling.

## Ephrin-B2 drives invasion and anchorage-independent proliferation of human mesenchymal GSC

We have shown that ephrin-B2 drives two key aspects of tumour formation in our murine models: GSC perivascular invasion and proliferation. We thus sought to determine the relevance of these findings to human GBM (hGBM). To this end, we first examined Ephrin-B2 levels at the infiltrative margin of 10 GBM patient specimens and correlated them with GSC marker expression (*Figure 6a*, *Figure 6—figure supplement 1a* and  *Table 3*). Two of these tumours were the original lesions from which G19 and G26 human GSC lines used below have been isolated. Serial sections were stained with H&E and with antibodies against Ephrin-B2 and ALDH1, a stem cell marker which labels GSC in perivascular and hypoxic niches (*Rasper et al., 2010*). All tumours presented cytoplasmic and membranous Ephrin-B2 expression in 25 to 90% of the tumour cells, as well as neurons, inflammatory cells and vascular endothelial cells, as reported (*Gale et al., 2001*; *Sawamiphak et al., 2010*; *Ottone et al., 2014*). In contrast, ALDH1 expression was less abundant, with expression detected predominantly in neoplastic cells surrounding blood vessels, in perinecrotic regions and at the infiltrative tumour margin, as expected from a stem cell marker (*Rasper et al., 2010*). Weaker ALDH1 expression was also detected in reactive astrocytes and endothelial cells. Importantly, many perivascular ALDH1$^+$ neoplastic cells at the infiltrative tumour margin co-expressed ephrin-B2. Thus, ephrin-B2 is expressed in stem-like cells invading along blood vessels in primary human tumours.

We next assessed *EFNB2* expression in a panel of 8 well-characterised primary human GSC lines isolated from independent patient tumours, including two of the tumours analysed by IHC above (*Caren et al., 2015*). These lines recapitulate the transcriptional sybtypes of primary GBM and predominantly cluster into proneural and mesenchymal subtypes, as previously reported for GSC (*Figure 6 b* and *Figure 6—figure supplement 1b*) (*Bhat et al., 2013*; *Mao et al., 2013*). Consistent with our mouse models, RNA-sequencing analysis revealed frequent upregulation of *EFNB2* in GSCs and a significant correlation between *EFNB2* and mesenchymal gene expression levels (*Figure 6b*). In addition, analysis of 402 GBM from the TGCA dataset classified according to Verhaak et al. and corrected for *IDH1* status, indicated that *EFNB2* expression levels are highest in mesenchymal and classical GBM subtypes (*Figure 6—figure supplement 1c*) (*Verhaak et al., 2010*). When tumours were divided into two groups on the basis of *EFNB2* levels relative to median expression within subtypes, *EFNB2* correlated with decreased survival in mesenchymal GBM (*Figure 6—figure supplement 1d*). Together, these results are indicative of a functional role for ephrin-B2 in GSC tumorigenesis within human GBM, specifically of mesenchymal subtype.

To test this more directly, we introduced control scrambled shRNAs or shRNAs to *EFNB2* in 3 of the mesenchymal GSC lines described above (*Figure 6—figure supplement 1e*) and assessed effects of Ephrin-B2 depletion on invasion and proliferation in vitro. As shown in *Figure 6c–h*, we found that, similar to the murine GSC models, in the absence of ephrin-B2, GSC (but not SCR shRNA-transduced controls) lost the ability to migrate over coated ephrin-B2 and failed to proliferate anchorage-independently, resulting in a G2/M cell-cycle arrest. These effects were not caused by an impairment of GSC self-renewal following ephrin-B2 downregulation and were highly specific, as overexpression of an shRNA-resistant *EFNB2* construct in G26 sh*EFNB2* cells fully rescued their anchorage-independent proliferation (*Figure 6—figure supplement 1f–h*).

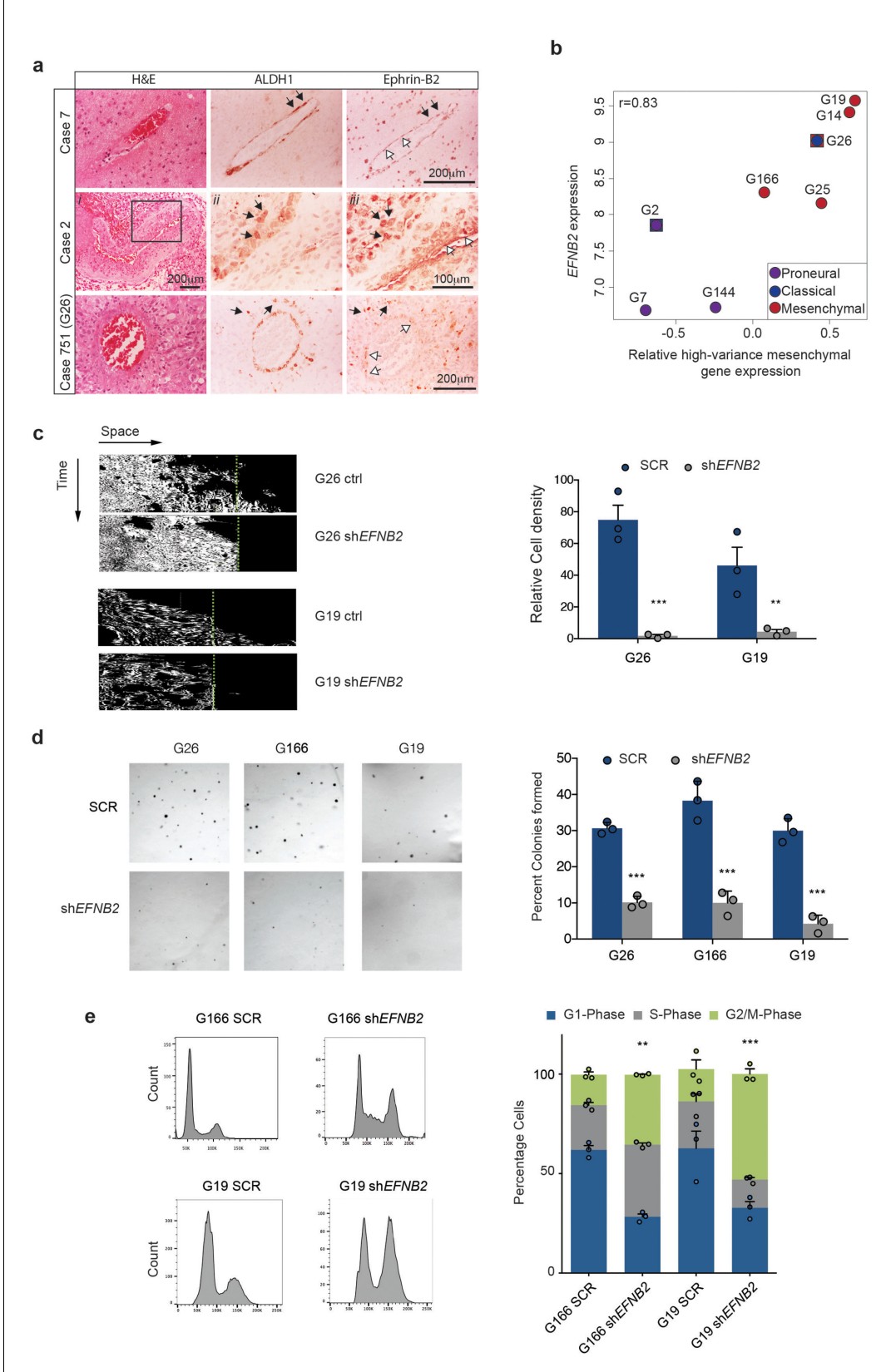

**Figure 6.** Ephrin-B2 is expressed in human GSC and drives their invasion and proliferation in vitro. (**a**) Haematoxylin-eosin (H&E), ALDH1 and Ephrin-B2 HRP staining of infiltrative margins of patient tumours. Three different glioblastomas of 10 analysed are shown. Atypical tumour cells expressing both

*Figure 6 continued on next page*

*Figure 6 continued*

ALDH1 and EphrinB2 (black arrows) can be identified around both normal and fibrotic co-opted vessels (white arrows) in all cases. Images *ii* and *iii* for Case 2 are close ups of the artery in *i*, showing that ALDH1+/EphrinB2+ cells present irregular and hyperchromatic nuclei characteristic of tumour cells. Box denotes magnified area. Case 751 is the original tumour from which G26 cells have been isolated. (**b**) Correlation plot between *EFNB2* and mesenchymal gene expression levels (Z-score) in 8 primary human GSC lines. Mesenchymal Z-score was calculated using signature marker genes with a variance greater than 0.1 (high-variance genes) across GSC lines. Verhaak classification of each line into disease subtype using the same cutoff is also shown. Of note, G2 and G26 display dual signatures as indicated. G26 carries a deletion in the *NF1* gene. (**c**) Representative kymographs (left) and quantifications (right) of human primary GSC lines G26 and G19, stably expressing scrambled shRNA (ctrl,) or shRNA directed against *EFNB2* (sh*EFNB2*). Note that while migration of human GSC is less pronounced than in mouse cells, *EFNB2* depletion results in complete inhibition of migration. Error bars denote s.e.m., n = 3. Students t-test. Green dots denote boundary of ephrin-B2 coating identified by fluorescence. (**d**) Left: Representative micrographs of SCR or sh*EFNB2* transduced G26, G166 and G19 cells cultured in soft agar for 14d. Right: quantification of number of colonies formed in soft agar in all cultures, expressed as percentage over total number of seeded cells. n = 3, error bars depict s.e.m. One way ANOVA with Tukey correction. (**e**) Left: representative FACS profiles of G166 and G19 cells ctrl or sh*EFNB2* grown in soft agar for 72 hr, showing DNA content by propidium iodide (PI) staining. Right: quantification of cell cycle phases from the FACS profiles. n = 3 as indicated by the dots. Error bars depict s.e. m., One way ANOVA with Tukey post hoc test shown for G2/M phase. See also *Figure 6—figure supplement 1* and *Figure 6—source data 1*.

The following source data and figure supplements are available for figure 6:

**Source data 1.** Raw data for quantifications of kymographs, number of colonies formed in soft agar and cell-cycle analysis of human GSC presented in *Figure 6*.

**Figure supplement 1.** Ephrinb2 expression is increased and correlates inversely with patient survival in mesenchymal human GBM.

**Figure supplement 1—source data 1.** Raw data for all quantitative analyses shown in *Figure 6—figure supplement 1*.

## *EFNB2* silencing or anti-ephrinB2 scFv Ab treatment suppress human GSC tumourigenesis

To test effects of ephrin-B2 on hGSC tumourigenicity in vivo , we used two complementary strategies. First, we xenotransplanted luciferase-tagged SCR or sh*EFNB2* G26 cells intracranially in nude mice and monitored tumour growth and disease-free survival by bioluminescence imaging and survival analysis (*Figure 7a–c*). We found a dramatic impairment of tumour growth in the *EFNB2* knock-down group relative to control, in that vector-transduced G26 formed GBM in all animals (as previously reported (*Stricker et al., 2012*) whilst *EFNB2* knock-down cells gave rise to slower growing tumours in only 2 out of 10 animals, with the other 8 mice remaining tumour-free for >1 year. To investigate the mechanisms responsible and confirm the generality of these findings, we repeated these experiments using both G26 and G19 lines and examined vascular association and proliferation of the GSC by immunofluorescence and FACS analysis at 10 days following implantation, a time point at which knock-down cells could still be detected by bioluminescence (*Figure 7b*). SCR GSC associated with the vasculature at this early time point (and at later stages of tumour growth), resulting in vascular co-option, as reported (*Figure 7d* and *Figure 7—figure supplement 1*) (*Watkins et al., 2014*). In contrast, vascular contact was severely compromised in knock-down cells.

**Table 3.** Patient information for tumours 1–8 used for Ephrin-B2 IHC

| name | gender | age | site | IDH1/2 | MGMT | ATRX | Grade |
|------|--------|-----|------|--------|------|------|-------|
| case 1 | M | 69 | left temporal | wt | 0 | retained | IV |
| case2 | M | 45 | left temporal | wt | 0 | retained | IV |
| case 3 | M | 54 | right temporo-parietal | wt | <10% | retained | IV |
| case 4 | M | 55 | right temporal | wt | 0 | retained | IV |
| case 5 | F | 61 | right parietal | wt | 25% | retained | IV |
| case 6 | M | 57 | right frontal | G395A | 0 | lost | IV (secondary) |
| case 7 | F | 73 | left frontal / crossed midline | wt | 0 | retained | Iv |
| case 8 | F | 39 | left temporo-parietal | wt | 0 | retained | IV |

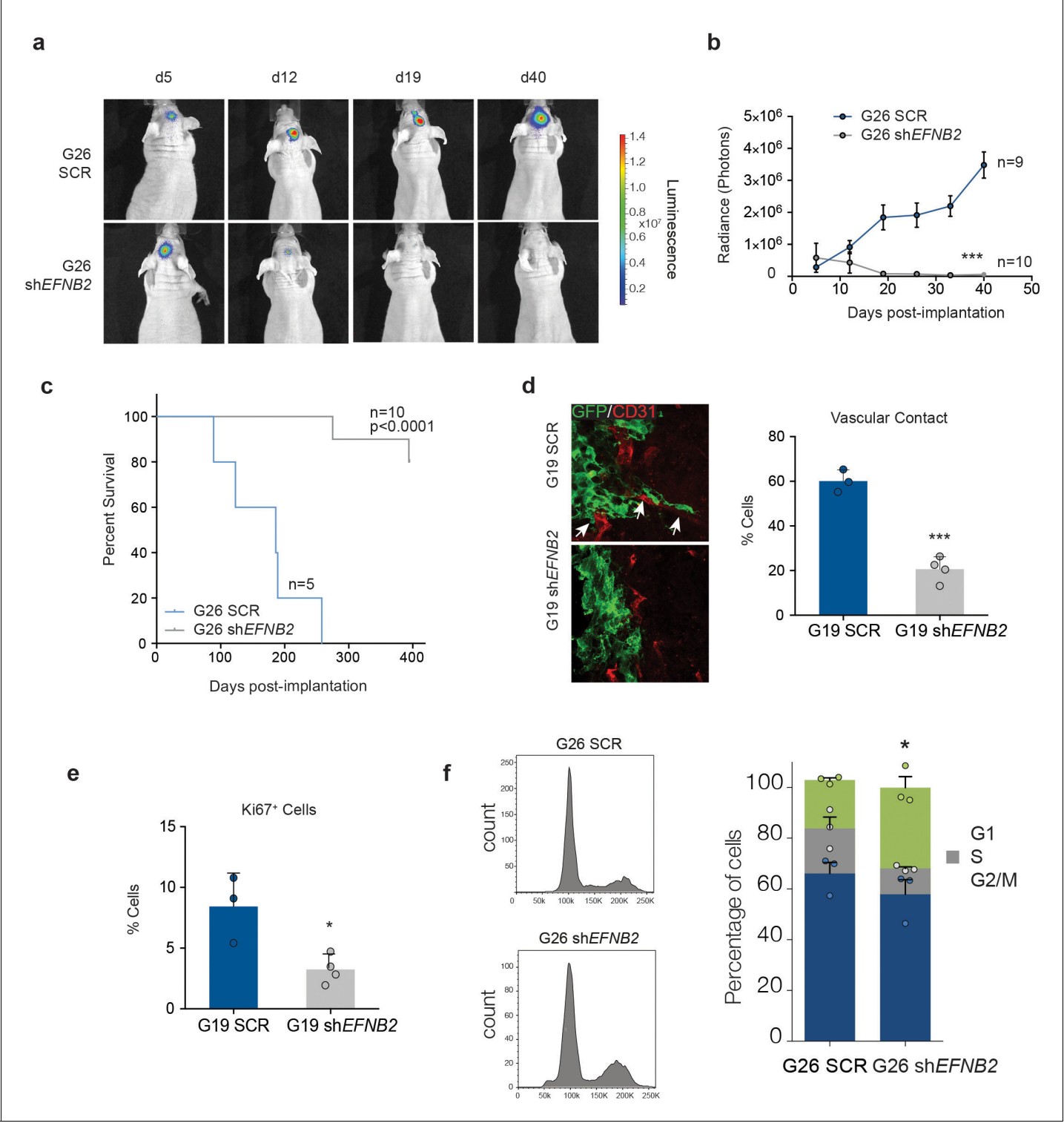

**Figure 7.** *EFNB2* silencing abolishes hGSC tumorigenicity. (**a**) Representative bioluminescent images of nude mice injected at d0 with $10^5$ luciferase-labeled G26 cells, stably expressing scrambled shRNA (SCR, top) or shRNA directed against *EFNB2* (sh*EFNB2*, bottom). (**b**) Quantification of luciferase bioluminiscence measured at the indicated time points in both groups. n = 9 for ctrl and 10 for sh*EFNB2*, error bars denote s.e.m. Two-way ANOVA with Tukey correction. (**c**) Kaplan-Meier survival plots of the mice depicted in (**a**). n = 5 for SCR, 10 for sh*EFNB2*. Log Rank Mantel Cox test. (**d**) Representative fluorescence images (left) and quantification (right) of GSC/vascular interactions in tumors derived from G19 SCR and G19 sh*EFNB2* cells at 10 days post-implantation. Sections were stained for GFP to identify tumor cells and CD31 (red) to label pre-existing blood vessels. Arrows indicate vascular association and co-option in SCR tumours, which is reduced in sh*EFNB2* tumours. (**e**) Quantification of percentages of Ki67+ cells over

*Figure 7 continued on next page*

*Figure 7 continued*

total GFP$^+$ cells in G19 SCR and sh*EFNB2* tumors. n = 4, error bars denote s.e.m. Students t-test. (**f**) PI FACS plots (left) and quantifications (right) of cell cycle profiles of G26 SCR and sh*EFNB2* cells retrieved from brain tissue 10 days after intracranial injection. n = 3 error bar denotes s.e.m. Significance is given for G2/M phase, one way ANOVA with Tukey post hoc test. See also *Figure 7—figure supplement 1* and *Figure 7—source data 1*.

The following source data and figure supplement are available for figure 7:

**Source data 1.** Raw data for quantifications of tumour growth by bioluminescence analysis, survival by Kaplan-Meier analysis, tumour cell intractions with the vasculature, Ki67 labelling and cell-cycle analysis of human GSC-derived tumours presented in *Figure 7*.

**Figure supplement 1.** G26 cells associate with blood vessels.

Furthermore, similar to GSC1*Efnb2*$^{-/-}$ murine cells, *EFNB2* downregulation resulted in a marked decrease in proliferation, accompanied by an arrest in the G2/M phase of the cell-cycle, indicative of a cytokinesis block (*Figure 7e,f*). Thus, Ephrin-B2 drives tumour initiation by mediating vascular association and proliferation of human GSC.

Second, we asked whether Ephrin-B2 inhibition could suppress tumourigenesis of pre-established tumours. For this, we took advantage of an Ephrin-B2 blocking scFv antibody fragment (B11) we previously developed (*Abengozar et al., 2012*). Migration and methylcellulose assays in vitro confirmed that B11 effectively inhibits Ephrin-B2 signalling in GSC1 (*Figure 8—figure supplement 1a,b*). Luciferase-tagged G26 cells were implanted intracranially in immunocompromised mice and 13d later, once sizeable, exponentially growing tumours had formed, but prior to the onset of neoangiogenesis, B11 was administered intravenously for a total of 9 consecutive days, as reported (*Abengozar et al., 2012*; *Binda et al., 2012*). This treatment regimen enabled us to assess direct effects of B11 on the tumour cells in the absence of confounding anti-angiogenic effects known to result from ephrin-B2 inhibition (*Sawamiphak et al., 2010*; *Abengozar et al., 2012*). Efficient delivery of the scFv across the blood/tumour barrier to the tumour cells was confirmed in parallel animals using Alexa-680 labelled B11 (*Figure 8—figure supplement 1c*). Remarkably, B11 strongly suppressed tumour growth in all animals without any evidence of toxicity, with three of six animals showing complete regression, as judged by quantitative bioluminescence imaging, survival analysis and post-mortem examination of the injected brains (*Figure 8a–c*). Analysis and quantification of the tumours immediately following treatment also revealed that, in contrast to vehicle-treated control tumours, invading tumour cells failed to associate with and coopt the vasculature (*Figure 8d*). Furthermore, proliferation was reduced in B11 tumours relative to controls, with B11 samples containing many Ki67$^+$ multinucleated cells, indicative of a cytokinesis defect (*Figure 8e*). Therefore, B11 suppresses G26 tumourigenicity through inhibition of ephrin-B2-dependent perivascular invasion and anchorage-independent proliferation, independent of angiogenesis. We conclude that Ephrin-B2 plays an important role in the pathogenesis of human GBM and its inhibition might be an effective strategy for curtailing GBM progression and recurrence.

## Discussion

Perivascular invasion is a critical mechanism of GBM growth and infiltration, which greatly contributes to the marked therapeutic resistance of these tumours (*Cuddapah et al., 2014*; *Scherer, 1938*). Here, using a novel mouse 'progression series' that mimics transformation of normal NSC to mesenchymal GSC, we identified ephrin-B2 as a critical driver of GSC perivascular invasion. Interestingly, a role for ephrin-B2 and its phosphorylation in the tumour cells has been previously linked to GBM invasiveness, albeit not in the context of GSC. This suggests that ephrin-B2 might promote invasion both by generally enhancing cell-intrinsic invasive potential through reverse signalling (*Nakada et al., 2010*; *Tu et al., 2012*) and, also, by specifically enabling perivascular spread through forward signalling as we have shown here.

We found that ephrin-B2 expressed on vascular endothelial cells compartmentalises non-transformed cells. A similar role for Eph/ephrin signalling in constraining migration of premalignant cells was reported in colorectal cancer, where ephrin-B ligands in the surrounding normal tissue inhibit invasion of incipient lesions through activation of EphB receptors in the tumour cells (*Cortina et al.,*

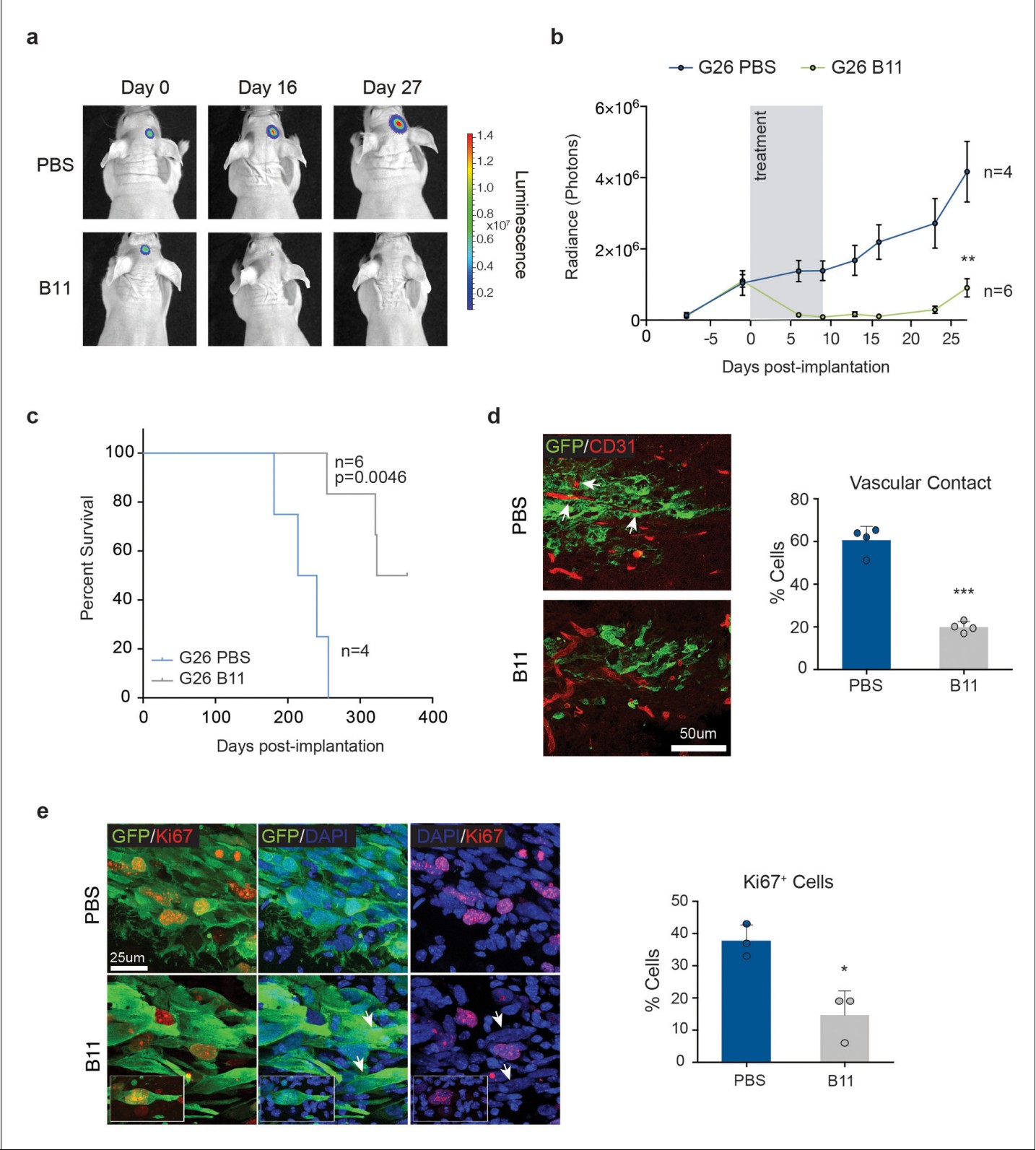

**Figure 8.** Treatment with anti-EphrinB2 ScFv blocking antibodies suppresses progression of established GBMs. (**a,b**) Representative images (**a**) and quantification (**b**) of bioluminescence radiance of PBS or anti-Ephrin-B2 scFv antibody (B11) injected mice. n = 4 for PBS ctrl and 6 for B11 treatment groups. (**c**) Kaplan-Meier survival plots of the mice depicted in a and b. n = 4 for PBS-treated, 6 for B11-treated G26 tumours. Log Rank Mantel Cox test. (**d**) Representative immunofluorescence images (left) of PBS and B11-treated G26 tumours 5 days after the first B11 injection, stained for GFP

*Figure 8 continued on next page*

*Figure 8 continued*

(green) to identify tumour cells and CD31 (red) to label the endogenous vasculature. Quantification of vascular association is shown on the right. Arrows indicate co-opted blood vessels in PBS-treated tumours. Scale bar 50 µm. Error bars denote s.e.m. Student t-test. (**e**) Representative immunofluorescence images (left) PBS and B11-treated G26 tumours stained for GFP and the proliferation marker Ki67. Quantification of the percentage of Ki67$^+$ cells over total number of GFP$^+$ tumour cells is shown on the right. Note the presence of multinucleated cells (arrows and inset) in B11-treated tumours. Scale bar=25 µm, n = 3, error bars denote s.e.m. Students t-test. See also *Figure 8—figure supplement 1* and *Figure 8—source data 1*.

The following source data and figure supplements are available for figure 8:

**Source data 1.** Raw data for quantifications of tumour growth by bioluminescence analysis, survival by Kaplan-Meier analysis, tumour cell intractions with the vasculature and Ki67 labelling of human GSC-derived tumours presented in *Figure 8*.

**Figure supplement 1.** B11 crosses the blood/tumour barrier and inhibits Ephrin-B2 on tumour cells.

**Figure supplement 1—source data 1.** Raw data for all quantitative analyses shown in *Figure 8—figure supplement 1*.

*2007*). Our study is also consistent with work by Astin et al., which reported that ephrin ligands on stromal cells repel normal and non-metastatic prostate cancer cells (*Astin et al., 2010*). Thus, the present work further strengthens the notion that ephrins expressed in normal tissues act as tumour-suppressors during early tumourigenesis by promoting tumour confinement and identifies the vasculature as a critical mediator of these effects. It is tempting to speculate that ephrin-B2-mediated vascular compartmentalisation might be a more general tumour-suppressive mechanism, limiting the spread of premalignant lesions across many cancer types.

In contrast, malignant transformation overrides vascular compartmentalisation to enable stereotypical GBM invasion along the perivascular space (*Baker et al., 2014*; *Farin et al., 2006*; *Watkins et al., 2014*; *Winkler et al., 2009*). Mechanistically, we found this to be dependent on the upregulation of ephrin-B2 in the GSC themselves, which elicits constitutive activation of Eph forward signalling in the tumour through homotypic cell-cell interactions. Consequently, this promotes perivascular spread in two ways. First, constitutive Eph activation increases repulsion between the tumour cells within the tumour bulk, thereby promoting GSC motility and scattering of single cells away from the tumour, a hallmark of GBM migration (*Vehlow and Cordes, 2013*). Second, it desensitizes the cells to extrinsic ephrin-B2 repulsion, thereby circumventing vascular confinement and permitting cells that have detached from the tumour mass to continue their migration along the vasculature (*Figure 8—figure supplement 1d*). Thus, invasive GSC augment their tumourigenic potential by 'hijacking' the same signalling pathway through which the vasculature suppresses tumourigenesis in normal tissues. Intriguingly, the mechanism of invasion we identified here, differs from what has been reported for colorectal and prostate cancer in the studies mentioned above (*Astin et al., 2010*; *Cortina et al., 2007*). Indeed, while in those systems tumour spread depended on the downregulation of Eph receptors in a cell-autonomous manner, in our GBM model non-cell-autonomous constitutive activation of Eph forward signalling is responsible, suggesting that the mechanisms that evade ephrin repulsion during tumour progression might be critically dependent on cancer type.

We further report that ephrin-B2 is also a critical effector of the 'Ras-transformed phenotype', capable of both transforming immortalised cells and mediating, at least in part, Ras-dependent transformation. Indeed, lgT-immortalised stem cells overexpressing *Efnb2* formed aggressive tumours with nearly identical kinetics to Ras-transformed cells and genetic deletion of *Efnb2* in GSCs significantly delayed tumourigenesis. We demonstrate that these effects depend on the surprising and previously unknown ability of ephrin-B2 to drive anchorage-independent proliferation through the activation of cytokinesis in a RhoA-dependent manner. It is of note that, in contrast to perivascular invasion, these ephrin-B2 effects are mediated by reverse signalling and independent of cell-cell contact, indicating that ephrin-B2 drives two key aspects of GBM tumourigenesis through a complex interplay of cell-autonomous and non-cell-autonomous mechanisms (*Figure 8—figure supplement 1d*). Previous studies in endothelial and smooth muscle cells have reported a similar cell-cell contact- and Eph-independent role for ephrin-B2 reverse signalling in driving acto-myosin contractility and cell spread, respectively (*Bochenek et al., 2010*; *Foo et al., 2006*). We currently do not know how

RhoA becomes activated downstream of ephrin-B2. A potential candidate is Dishevelled, which mediates activation of Rho/ROCK by ephrin-B1 (*Tanaka et al., 2003*). It would be interesting to explore the role of Dishevelled in our system in the future.

Our results in GBM specimens and primary human GSC lines strongly support a critical role for ephrin-B2 in the human disease through the same mechanisms that we identified in the murine models. Indeed, *EFNB2* knock-down prior to implantation abrogated tumour initiation and treatment of pre-existing intracranial tumours with Ephrin-B2 blocking antibodies, under conditions that mimic human therapeutic paradigms, strongly reduced the growth and expansion of pre-formed tumours. Significantly, both approaches curtailed tumour growth by impairing both vascular association and cytokinesis. In patient samples stratified by GBM subtype, *EFNB2* levels were highest in mesenchymal and classical tumours and correlated, inversely, with survival specifically in mesenchymal GBM. This is in agreement with our mesenchymal mouse models and analysis of human GSC lines, which revealed a robust correlation between *EFNB2* and mesenchymal gene expression levels. This recurrent correlation is particularly noteworthy, as *EFNB2* was previously identified as a component of the core mesenchymal gene network (*Carro et al., 2010*). Furthermore, increasing evidence suggests that convertion to the mesenchymal signature is associated to progression, radioresistance and recurrence and is thus a fundamental driver of GBM malignancy (*Ozawa et al., 2014*; *Bhat et al., 2013*).

GSCs are thought to be largely responsible for tumour infiltration and propagation (*Chen et al., 2012a*; *Venere et al., 2011*). As our work is based on murine and human GSC models of GBM, and Ephrin-B2 levels are high in the GSC compartment in primary tumours (*Figure 6a*), our findings indicate that anti-ephrin-B2-based therapies would target the most critical subset of cells within these lesions. As such, by harnessing the complexity of the Eph/ephrin system, inhibition of ephrin-B2 would in itself be an effective 'combinatorial therapy', capable of suppressing two critical GSC-intrinsic properties, namely perivascular invasion and proliferation, and might thus represent an attractive strategy for blocking GBM progression and recurrence. In addition, as ephrin-B2 levels are robustly increased in GSC compared to normal neural stem cells and other tissues, such therapies should be relatively tumour-specific and non-toxic.

## Materials and methods

### Animals

All animal work was carried out in accordance with the regulations of the Home Office and the ARRIVE guidelines. *Efnb2*$^{i\Delta EC}$ mice and recombination protocols were described previously (*Wang et al., 2010*; *Ozawa and James, 2010*; *Ottone et al., 2014*). Tamoxifen-injected *Efnb2*$^{fl/fl}$ littermates were used as controls. C57Bl6 and CD-1 nude mice for tumourigenicity analysis were obtained from Charles River. For GSC1 tumorigenic studies and B11 treatments of G26 cells, sample size was calculated by power analysis using PASS software based on pilot studies assessing the tumorigenicity of the parental line. For experiments with sh*EFNB2* human G26 cells sample size was set based on previous studies (*Pollard et al., 2009*).

### Craniotomies and intravital imaging

Craniotomies were performed as previously described (*Holtmaat et al., 2009*). Briefly 6–8 week old *Efnb2* $^{i\Delta EC}$ or *Efnb2*$^{fl/fl}$ mice were anaesthetized with ketamine-xylayzine intraperitoneal injection (0.083 mg/g ketamine, 0.0078 mg/g xylazine). The animals were then injected with 0.02 ml 4 mg/ml intramuscular dexamethasone to limit an inflammatory response and subcutanaeous bupivacaine (1 mg/kg) a local anaesthetic. Once the skull was exposed a few drops of lidocaine (1% solution) were applied on its surface. Before covering the burr hole with a glass coverslip, $2x10^4$ cells were injected into the cortex at a depth of 150–300 μm using a picospritzer. Mice were left to recover for 7 days and then imaged using a purpose built microscope equipped with a tunable Coherent Ti:Sapphire laser and PrairieView acquisition software. Mice were anaesthetized with isoflurane and secured to a fixed support under the microscope. The eyes were coated with Lacri-lube (Allergan) to prevent dehydration, an underlying heat pad used to maintain body temperature (37°C). To prevent dehydration isotonic saline solution was administered (i.p.) during long imaging sessions. Depth of anaesthesia was closely monitored. To visualize blood vessels a 3000MW dextran-Texas Red

conjugate was injected intravenously prior to imaging. A pulsed laser beam with a wavelength of 910 nm was used to ensure that both GFP-tumour cells and Texas Red showed sufficient signal intensity. Each imaging session lasted for no longer than 60 min and mice were imaged up to four times daily with cells imaged every 20 min . After image acquisition individual frames were aligned and the displacement of single cells measured using ImageJ software.

## Orthotopic xenografts and in vivo imaging

$5x10^4$ luciferase expressing cells were injected into 6–8 week old CD-1 nude mice into the right putamen (1 mm rostral to bregma, 2 mm lateral and 2.5 mm depth) as previously described (*Ozawa and James, 2010*). Tumour cells were loaded into the syringe just prior to injection and the needle kept in place for a further 5 min to ensure minimal reflux of the material along the needle tract. Tumour formation, growth and volume were indirectly calculated by sequential images taken with an IVIS Spectrum in vivo imaging system (Perkin Elmers). Following administration of 120 mg/kg D-luciferin (Intrace medical) by intraperitoneal injection, mice were anaesthetized (3% isoflurane) and imaged under continuous exposure to 2% isoflurane. Luminescent measures were performed once a week starting 5 days after cell implantation until day 40. Bioluminescence was detected by the IVIS camera system, integrated, digitised, and displayed. Pseudocolor scale bars were consistent for all images of dorsal views in order to show relative changes at tumour site over time. Tumours were quantified by calculating total flux (photons/s/cm$^2$) using Living Image software (Xenogen, Caliper Life Sciences).

Animals were sacrificed when they showed signs of distress or weight loss. For treatment experiments, PBS or B11 administration was started once the tumours reached a minimum signal intensity of $1x10^6$ photons/s/cm$^2$. Mice were then randomised into two groups prior to intravenous injection of 5 doses of either anti ephrinB2-scFv B11 (total dose 20 mg/kg) or PBS control over a period of 9 days. Survival curves were estimated using the Kaplan-Meier method. Significance was calculated using the log-rank Mantel-Cox test. To determine efficiency of delivery, B11 was labeled with Alexa Fluor 680 using the SAIVI rapid antibody kit (Invitrogen) according to the manufacturer's instructions.

## Cell culture, gene delivery and constructs

Primary mouse microvascular endothelial cells were obtained from Caltag Medsystems and subcultured according to the suppliers recommendations (cells were primary cells isolated directly from normal mouse brain, no further authentication performed by the authors, mycoplasma negative as tested by Mycoalert kit, Lonza)

Human GSC lines were described previously (*Pollard et al., 2009*; *Caren et al., 2015*). Cells were originally isolated from patient tumours and are maintained in serum free cultures on laminin (*Pollard et al., 2009*, no further authentication performed by the authors, mycoplasma negative as tested by Mycoalert kit, Lonza). Co-culture experiments with endothelial cells for assessment of p-Eph levels were performed as previously described (*Ottone et al., 2014*). Briefly, endothelial cells were seeded at confluence on PLL-coated dishes and imNSC or GSC cells seeded on top the following day. The cells were separated by differential trypsinisation. For analysis of clonal efficiency, single cells were sorted into individual wells of a low attachment 96 well plate using a FACS Aria III cell sorter and cultured in neural stem cell media for 7d after which percentage of neurosphere formed was calculated. Transient transfections and viral transductions were performed as previously reported (*Ottone et al., 2014*). The following plasmids were used: FUGW was used to label all imNSC and GSC cells (*Lois et al., 2002*). Constitutively active RhoA-V14 and dominant negative RhoA-N19 constructs were a kind gift of Anne Ridley (*Ridley and Hall, 1992*). Plasmid encoding full-length cDNAs of human EphB1 was purchased from OriGene Technologies, Inc (RC214301) and mouse EphB2 was a kind gift of E. Batlle.

## Generation of imNSC and GSC lines

Primary NSCs were isolated directly from the SVZ of postnatal or adult *Cdh5*(Pac)-CreERT2 or *NF1*$^{fl/fl}$ transgenic animals and cultured as described previously (*Wang et al., 2010*; *Zhu et al., 2001*; *Ottone et al., 2014*). To generate imNSC and GSC lines the following plasmids were used: pBabe-largeT + pLXSN-hRasV12 (*Cremona and Lloyd, 2009*), shp53 pLKO.1 (*Godar et al., 2008*), pCMV-

Cdk4 (*van den Heuvel and Harlow, 1993*) (addgene 1874). Recombination of the *NF1*^*fl/fl* allele was achieved using Adeno-Cre viruses at an MOI of 100, as reported (*Ottone et al., 2014*). All experiments were performed on a minimum of two independent batches of GSC1 and 2 generated as described above from independent primary preparations (no further authentication performed by the authors, mycoplasma negative as tested by Mycoalert kit, Lonza). Cells were used within the first 10–15 passages from infection.

## Immunoblotting, immunocytochemistry and immunohistochemistry

Western blots and immunocytochemistry were performed as reported previously (*Ottone et al., 2014*). The RhoA activation assay kit (Millipore) was used according to manufacturers instructions. For immunohistochemical analysis of tumours, mice were perfused with 4% PFA and the brain postfixed in 4% paraformaldehyde for 2 hr, placed in a 30% sucrose solution over night, embedded in OCT and snap frozen. Immunohistochemistry was performed on 30 µm cryostat sections. Tissue sections were stained overnight at 4°C with antibodies diluted in 10% goat serum. Hematoxylin and Eosin (H&E) staining was performed on 3 µm paraffin embedded sections. The following commercial primary antibodies were used: β-Tubulin (Sigma T8328 1:5000), BrdU (Roche 11170376001 1:400), cleaved caspase-3 (Cell Signaling #9664 1:500), EphA4 (abcam ab641 1:1000), EphB1 (abcam ab129103 1:1000), EphB2 (abcam ab76885 1:1000), EphB3 (abcam ab133742), EphB4 (R&D AF446), ephrinA5 (R&D AF3743 1:500), ephrinB2 (R&D AF496 1:250), CD31 (BD 550274 1:400), GAPDH (abcam ab9483 1:1000), GFAP (abcam54554 1:400), GFP (Invitrogen A-21311 1:400), Nestin (Millipore MAB353 1:400), pan-pEph (abcam ab61791 1:500), pFAK (Cell signalling #3283 1:500), p-Src (Cell Signaling 2101s 1:1000), O4 (R&D MAB1326 1:400), Sox2 (Cell signalling 3728s 1:250), SSEA1 (BD Pharmingen 560079 1:400), Tuj1 (Covance MMS-435P 1:500) and NG2 (Chemicon, 1:250).

## Migration assays and quantifications

For migration assays, endothelial cells and imNSC/GSCs were plated at a density of $2 \times 10^4$ cells into adjacent compartments of cell-culture silicon inserts (Ibidi) separated by a 500 µm gap. Alternatively one well of the insert was coated over night with 4 µg/ml of recombinant ephrin-B2-Fc fusion proteins or Fc controls (R&D) clustered at a molar ratio of 1:2 with fluorescently labelled anti-Fc antibody for 1.5 hr at room temperature. After removal of the insert the cells were cultured in medium supplemented with 1 % Matrigel (Invitrogen) and live cell imaging was performed in a heated and $CO_2$ controlled chamber for 60 hr. For stimulation of imNSCs, recombinant ephrinB2-Fc ligands were preclustered with anti-human Fc antibodies as above and added to imNSCs for 24 hr before removal of the insert at a final concentration of 10 µg/ml. Analysis of ephrinb2-stimulated imNSC migration was terminated at 48 hr to achieve maximal stimulation of forward signaling throughout the experiments. Migration was quantified by tracing the boundary between GFP positive NSCs and non-stained bmvECs in ImageJ. To analyse the number of cell contacts individual frames from the videos of cells making initial contact with the ephrinB2-Fc boundary were analysed. The total number of protrusions per cell that were in contact with other cells at the time of entering the ephrin-B2-coated well were counted. All counting was performed blind. Kymograph analysis was performed using an ImageJ macro. Quantification of the kymographs was performed by measuring pixel intensities 200 µm before and after the ephrinB2-Fc boundary at the last imaged time-point to assess the proportion of cells that migrate over ephrin-B2 upon contact (expressed as relative cell density). To analyse scattering behaviour, cells were seeded sparsely at 10,000 cells/12 well and their migration tracked for 20 hr . Collisions between single cells were quantified over 200 min and a minimum of 50 cells were counted per condition per biological repeat.

## Soft-agar and Methylcellulose cultures

For analysis of anchorage independent proliferation, cells were seeded for up to 72 hr in SVZ culture medium containing 1.8% dissolved Methylcellulose (Sigma) to form a semisolid hydrogel as previously described[47]. To retrieve cells, the suspension was diluted fivefold with DMEM and centrifuged at 500 g for 10 min. The pellet was then washed twice with ice-cold PBS. For analysis of binucleated cells, all cells were incubated with EdU (Life technology) for 4 hr and placed on PLL-coated coverslips 15 min before fixation and staining. As the metaphase in mammalian cells typically lasts less than an hour this experimental set up enabled us to distinguish between cycling (EdU positive) and

cytokinesis arrested (EdU negative) cells. To assess colony formation in soft agar $5 \times 10^3$ cells/6 well were seeded into 0.5% agar on top of a bottom layer of 1% agar (Sigma). Total number of colonies formed was counted after 10 days.

## Flow cytometry

Cell pellets retrieved from methylcellulose were resuspended, fixed in 70% EtOH for 4 hr and stained with propidium iodide (Sigma). A minimum of 10,000 cells were counted for each condition using a BD LSR II. The PI profile was then analysed using the cell-cycle module of FlowJo X. Cells from tumours were isolated as follows: The GFP positive tumour tissue was dissected using a fluorescent dissection microscope and digested using papain according to the manufactures recommendation (Worthington Biochemical Corporation). Cells were stained with PI as above prior to FACS analysis. All FACS stainings were performed on at least 3 independent cultures/tumours.

## RNA sequencing and Bioinformatic analysis

GFP-labeled GSC1 and GSC2 tumours were dissected and digested as above. RNA was isolated by using RNAeasy Plus Mini Kit (Qiagen) and RNA sequencing libraries were constructed using the NEBNext Ultra RNA Library Prep Kit for Illumina (NEB). RNA-seq data from adult brain of normal mice was obtained from GEO [1] (accession numbers GSM1055111, GSM1055112 and GSM1055113). Sequencing reads from tumour samples were aligned with the Tophat splice junction mapper (*Kim et al., 2013*), version 2.0.11 against GRCm37/ mm9 and transcript annotations. All parameters were set to default except inner distance between mate pairs (r = 100) and library type (fr-firststrand). The normal brain data was also aligned using Tophat with default parameter values except for distance between mature pairs (r = 200). The DESeq2 Bioconductor package (version 1.4.5) (*Love et al., 2014*) was used to perform differential gene expression analysis on read counts obtained with HTSeq-count (version 0.5.3p9) (*Anders et al., 2015*), and p-values were adjusted for multiple testing with the Benjamini-Hochberg procedure (*Love et al., 2014*). Genes with adjusted p-value <= 0.05 were deemed to be differentially expressed.

Sequencing reads for NS and GSC cell lines were aligned to mouse genome build GRCm38/ mm10 with STAR 2.5.2a (*Dobin et al., 2013*) using the two-pass method for novel splice detection (*Engstrom et al., 2012*). Read alignment was guided by gene annotation from Ensembl release 84 (*Cunningham et al., 2015*) with optimal splice junction donor/acceptor overlap settings. Transcripts were quantified with HTSeq-count (*Anders et al., 2015*) based on feature coordinates from Ensembl 84. Gene set enrichment analysis was carried out with the GAGE Bioconductor package (version 2.18.0) (*Luo et al., 2009*). Gene sets used corresponded to either biological process (BP) terms from the Gene Ontology, or derived from CNS cell type RNA-seq data described by the Barres group (http://web.stanford.edu/group/barres_lab/brainseq2/brainseq2.html) using mean log2 FPKM values for astrocyte, neuron, OPC and oligodendrocyte classes. The top 500 genes for each class were identified by the ratio of individual class average expression to the maximum non-class average expression values. Log2 fold-change values computed by DESeq2 (version 1.8.2) were used as the input for GSA analysis. Data visualised in *Figure 1—figure supplement 1a* is derived from log2 FPKM data for the left panel and variance stabilised read counts using the rlog function in DESeq2.

## GBM subtype analysis

Subtype analysis of human GSC lines was completed using methods and microarray data described by Caren et al. (*Caren et al., 2015*). Sample log2-transformed expression values for the signature centroid genes were produced by taking the mean expression across sample replicates. Centroid genes that could not be assigned to annotated genes were also omitted from further analysis. To ensure the accuracy of subtype expression estimates only subtype genes with high variance across the GNS dataset were carried forward. Subtype scores per sample were computed from mean Z-score transformed levels of overexpressed centroid genes for each subtype. Samples were then classified as belonging to the subtype associated with the highest mean Z-score or mixed subtype when presenting a similarly high expression of another subtype's mean Z-score.

Gene set enrichment analysis (GSEA) (*Subramanian et al., 2005*) was applied to mouse tumor samples to test for Verhaak et al. subtype enrichment using the pre-ranked GSEA tool. DESeq2 computed T statistics from each tumor to control mouse sample differential expression results were used

to rank the gene lists. Pre-defined genesets pertaining to Verhaak et al. (*Verhaak et al., 2010*) were obtained from the Molecular Signatures Database. TCGA data analysis was performed as described previously (*Binda et al., 2012*) survival and expression analysis was performed on the publicly available TCGA data and relative mRNA expression obtained from the TCGA data portal (http://cancer-genome.nih.gov/dataportal/data/about). The analysis was performed using samr R package. A two class unpaired model was used with the following parameters: delta 0.01, number of permutation 100.

## Quantitative RT-PCR and siRNA/shRNA knockdown

Quantitative RT-PCR analysis and siRNA transfections were performed as previously described (*Ottone et al., 2014*). See *Supplementary file 3* for primer details. For knockdown of *EFNB2* in human G26 cells, lentiviral shRNA constructs were used (Invitrogen). Control cells were infected with control vectors. Both cell types were selected with puromycin and knock-down efficiency validated by qPCR and immunoblotting prior to tumourigenic studies.

## Neuropathological assessment of Ephrin-B2 expression in human GBMs

EphrinB2 expression pattern was investigated in brain tissue samples of eight patients operated of supratentorial hemispheric glioblastoma at Imperial College Healthcare Trust between February and June 2015. None of patients had radiotherapy or chemotherapy prior to surgery or any relevant comorbidities. Patients consent was given for all samples. The project received ethical approval from the Imperial College Healthcare Tissue Bank committee on behalf of MREC (ICHTB HTA licence: 12275, REC Wales approval: 12/WA/0196). All 8 tumours were wild type for IDH1 and 2 mutations and ATRX status with the exception of case 6, which was a secondary glioblastoma bearing the common G395A amino acid substitution in IDH1 and showing loss of ATRX expression.

Two additional GBM samples were the original lesions from which G19 and G26 cells have been isolated (*Pollard et al., 2009*). Preoperative structural MRI was available in all patients; the T2-weighted FLAIR sequences were reviewed to assess the extent of invasion into normal tissue. All patients underwent maximally safe surgical debulking and the isolated tissues were extensively sampled in order to represent the bulk of the tumour along with its infiltrative component in the surrounding grey matter. Specifically, areas representative of the invasive front of the tumour were chosen from each case. The tissues were fixed in 4% buffered formalin for 24 hr and then processed using standard method to produce paraffin sections. Sequential three-micron sections cut from each selected block and were used for immunoperoxidase immunohistochemistry. Immunostainings were performed following a standard protocol. Briefly, the sections were dewaxed in xylene and rehydrated in decreasing alcohols to distilled water. Antigen retrieval was performed incubating the sections for 20 min in steam-heated sodium citrate buffer (10 mM Sodium Citrate, 0.05% Tween 20, pH 6) at 90°C or 1 mM EDTA at pH 8. Following antigen unmasking, endogenous peroxidase was quenched with 1% hydrogen peroxide in PBS at -20°C for 15 min. After rinsing in PBS, the sections were incubated overnight at room temperature with the following primary antibodies: anti-EphrinB2 (clone EFR-163M hybridoma supernatant 1:4, CNIO) (*Abengozar et al., 2012*), anti-ALDH1 (monoclonal, BD Bioscience, dilution 1:500), anti-ALDH1 (mouse, clone 44, BD biosciences). The ephrin-B2 antibody gave robust membrane and cytoplasmic staining and labelled the endothelium, confirming specificity. Nuclear labelling was however detected in some tumour cases, a feature not present in control samples. Only membranous and cytoplasmic staining was scored as positive in the analysis of the tumours. The SuperSentitive IHC detection system from BioGenex (Fremon, CA, USA) was used to visualise antibody binding following the manufacturer's instructions and counterstained with Mayer's Haemalum.

## Statistics

Statistical analysis was performed using GraphPad Prism statistical analysis software. The precise tests are stated in the figure legends. Shapiro-Wilk test was used to confirm normal distribution of the data. All experiments for which quantifications were performed were carried out a minimum of three times as indicated in the figure legends.

## Acknowledgements

This work was funded by the Medical Research Council, UK (CO, BK, MPC, KG, SK, AA, TD and SP), the Royal Society (SP) and the Regional Government of Madrid/European Social Fund (JLM-T) We thank A Uren and C Jorgensen for critical reading of the manuscript, A Lloyd, A Ridley, E Batlle, H Augustin and M Herlyn for constructs and A. Lloyd for NF1$^{fl/fl}$ brain tissue. We are also grateful to the Imperial College Healthcare Tissue Bank for supplying the human glioblastoma samples and to the patients who donated their tissue for research.

## Additional information

### Funding

| Funder | Grant reference number | Author |
|---|---|---|
| Medical Research Council | Cell Interactions and Cancer, MC_AS A652 5PZ10 | Benjamin Krusche<br>Cristina Ottone<br>Melanie P Clements<br>Katrin Goetsch<br>Sanjay Khadayate<br>Azhaar Ashraf<br>Timothy Davies<br>Simona Parrinello |
| Regional Government of Madrid | European Social Fund | Jorge Martinez-Torrecuadrada |
| The Royal Society | RG110360 | Simona Parrinello |

The funders had no role in study design, data collection and interpretation, or the decision to submit the work for publication.

### Author contributions

BK, SP, Conception and design, Acquisition of data, Analysis and interpretation of data, Drafting or revising the article; CO, TD, Acquisition of data, Analysis and interpretation of data, Drafting or revising the article; MPC, KG, HL, AA, VDP, Acquisition of data, Drafting or revising the article; ERJ, Conception and design, Analysis and interpretation of data, Drafting or revising the article; SGM, PS, Acquisition of data, Drafting or revising the article, Contributed unpublished essential data or reagents; SK, Analysis and interpretation of data, Drafting or revising the article; SMP, Conception and design, Drafting or revising the article, Contributed unpublished essential data or reagents; FR, PB, Conception and design, Analysis and interpretation of data, Drafting or revising the article, Contributed unpublished essential data or reagents; JM-T, Conception and design, Acquisition of data, Drafting or revising the article, Contributed unpublished essential data or reagents

### Author ORCIDs

Ewan R Johnstone, http://orcid.org/0000-0002-6656-6088
Simona Parrinello, http://orcid.org/0000-0003-0820-6292

### Ethics

Animal experimentation: All animal work was carried out in accordance with the regulations of the Home Office and the ARRIVE guidelines and performed under the authority of Project License 70-7292. Patients consent was given for all human GBM samples. The project received ethical approval from the Imperial College Healthcare Tissue Bank committee on behalf of MREC (ICHTB HTA licence: 12275, REC Wales approval: 12/WA/0196).

## Additional files

### Supplementary files

• Supplementary file 1. List of significantly enriched GO terms in GSC vs NSC listing FDR q-values

• Spplementary file 2. List of significantly enriched GO terms in NSC vs GSC listing FDR q-values
• Supplementary file 3. Mouse Primers used for qRT-PCR

### Major datasets

The following previously published datasets were used:

| Author(s) | Year | Dataset title | Dataset URL | Database, license, and accessibility information |
|---|---|---|---|---|
| Paul Bertone | 2015 | Expression profiling of normal and glioblastoma-derived neural stem cells | http://www.ebi.ac.uk/arrayexpress/experiments/E-MTAB-3864/ | Publicly available at EBI ArrayExpress (accession no: E-MTAB-3864) |
| Alexey Fushan | 2015 | Gene Expression Defines Natural Changes in Mammalian Lifespan | http://www.ncbi.nlm.nih.gov/geo/query/acc.cgi?acc=GSM1055111 | Publicly available at NCBI Gene Expression Omnibus (accession no: GSM1055111) |
| Alexey Fushan | 2015 | Gene Expression Defines Natural Changes in Mammalian Lifespan | http://www.ncbi.nlm.nih.gov/geo/query/acc.cgi?acc=GSM1055112 | Publicly available at NCBI Gene Expression Omnibus (accession no: GSM1055112) |
| Alexey Fushan | 2015 | Gene Expression Defines Natural Changes in Mammalian Lifespan | http://www.ncbi.nlm.nih.gov/geo/query/acc.cgi?acc=GSM1055113 | Publicly available at NCBI Gene Expression Omnibus (accession no: GSM1055113) |

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
