## [Decision Letter]

Thank you for submitting your article "EphrinB2 drives perivascular invasion and proliferation of glioblastoma stem-like cells" for consideration by *eLife*. Your article has been reviewed by two peer reviewers, and the evaluation has been overseen by a Reviewing Editor and Sean Morrison as the Senior Editor.

The reviewers have discussed the reviews with one another and the Reviewing Editor has drafted this decision to help you prepare a revised submission.

We are interested in the paper if you can make text revisions that address the referee concerns. The first concern of Reviewer #1 is particularly important for two reasons: 1) knowing which cell type the transformed cells are is of great importance. One cannot assume that transformed NSCs remain NSCs. 2) it only takes a simple profiling experiment followed by GSEA, which shouldn't take long at all. This small addition would fix a major hole and greatly increase the impact of this paper.

*Reviewer #1:*

This manuscript thoroughly examined the role of ephrin-B2 in GBM stem cells. Using two complementary methods, SV40-T antigen + RasV12; or NF1-KO + p53-KD + CDK4 over-expression, the authors transformed NSCs into GBM stem cells (GSCs). After grafting into syngeneic mice, it was found that while GSCs migrated along blood vessels, control immortalized NSCs (imNSCs) failed to do so. Then it was shown that ephrin-B2 is sufficient and necessary for endothelial cells (EC) to repel imNSCs, using an in vitro compartmentalization assay. To investigate why GSCs were not repelled by vasculature, the expression levels of all Ephs and ephrins were examined in GSCs vs. imNSCs. It was found that ephrin-B2 is over-expressed in GSCs, and knockout of which led to repulsion by endothelial cells in vitro and loss of perivascular migration in vivo. Furthermore, over-expression of ephrin-B2 in imNSCs conferred them migratory activity over ECs in vitro and in vivo. Next it was shown that homotypic cell-cell interactions among GSCs helped them overcome the repulsion by endothelial cells, through constitutively elevated forward signaling to desensitize Eph receptors to exogenous stimulation. Pre-stimulation of imNSCs with ephrin-B2-Fc allowed them to overcome EC repulsion, further supporting this notion. Next it was shown that reverse signaling of ephrin-B2 also plays an important role in malignancy. Mechanistic studies showed that ephrin-B2 over expression led to anchorage-independent cytokinesis through RhoA-mediated reverse signaling. Finally, the human relevance of the study was demonstrated with data showing that both ephrin B2 knockdown and anti-ephrin-B2-scFv antibody treatment suppressed human GSC tumorigenesis in grafted mice.

Overall the experiments in this manuscript are meticulously designed with ample n number and proper controls. Most conclusions are supported with both gain-of-function and loss-of-function evidences. The most impressive part is the careful dissection of the roles of forward and reverse signaling of ephrin-B2 in perivascular migration of glioma cells and the G2/M cell cycle progression, respectively. The data with human glioma cells are very impressive as well.

The main concern is the rigor in glioma model establishment.

1) It has been shown in the literature that NF1-mutated NSCs could divert their fates into glial cells [PMID 22901811]. Therefore, it would be important to thoroughly examine the gene expression profile in comparison to all glial cell types (astrocytes, oligodendrocytes, and oligodendrocyte precursor cells, aka NG2 cells), rather than to call them stem cells based on a few marker genes (nestin, sox2, CD133, and SSEA1).

2) The methods to generate "GBM stem cells" by in vitro introduction of multiple mutations into NSCs are less than ideal. These cells could lose homeostatic contact inhibition thus proliferate uncontrollably rather than bona fide tumor cells. Since the premise of the paper is based on these cells, all the beautiful data could be artificial rather than biological. Of course, data with human GBM cells greatly alleviate this concern. Still, it would be wise for authors to tone down the glioma claim with these in vitro transformed mouse NSCs.

*Reviewer #2:*

In the manuscript presented Krusche et al. examine the mechanisms which lead to tumorigenesis by glioblastoma stem-like cells (GSCs). To accomplish this the authors use two models in which they can generate GSCs by RTK activation and p53/RB inactivation, and compared the properties of these cells to similar neural stem cells. Interestingly, they demonstrate that the GSCs, but not NSCs, are able to drive tumorigenesis, and they show that upregulation of ephrinb2 in these cells causes this tumorigenesis by two distinct processes: increase in cell proliferation and increasing perivascular invasion. Remarkably, these two processes are regulated by distinct ephrin-B2 mediated molecular mechanisms. Cell proliferation is mediated by reverse ephrin-B2 signaling by activation of RhoA whereas perivascular invasion is regulated by forward ephrin-B2 signaling onto neighboring cells causing desensitization to this pathway. This manuscript represents a tour de force in our understanding of both cell intrinsic and extrinsic mechanisms of malignant cell progression. The authors use elegant in vitro and in vivo models and demonstrate both the necessity and the sufficiency of overexpression of ephrin-B2 in the ability of these cells to proliferate and invade. This provides a significant advancement in our understanding of how tumor cells proliferate and interact with their surroundings.

---

## [Author Response]

We performed RNA-seq profiling experiments of cultured GSC1 and GSC2, as requested. As shown in new Figure 1—figure supplement 1 and tables 1-4, the analysis indicated that neither line undergoes differentiation into glial lineages. This further corroborates our previous findings (Figure 1—figure supplement 1) that the transformed cells retain both high clonal efficiency and the ability to differentiate into all three neuronal lineages. Therefore, these combined experiments eliminate any concerns that the transformed NSC lines may not retain NSC properties.

*1) It has been shown in the literature that NF1-mutated NSCs could divert their fates into glial cells [PMID 22901811]. Therefore, it would be important to thoroughly examine the gene expression profile in comparison to all glial cell types (astrocytes, oligodendrocytes, and oligodendrocyte precursor cells, aka NG2 cells), rather than to call them stem cells based on a few marker genes (nestin, sox2, CD133, and SSEA1).*

We thank the reviewer for raising this valid point. The immunofluorescence analysis initially presented in old Figure 1—figure supplement 1 has now been replaced with RNA-seq profiling of the GSC lines in vitro (New Figure 1—figure supplement 1 and tables 1-4). The analysis shows that neither GSC line differentiates aberrantly into glial cells as a result of elevated Ras signalling, thus confirming that both models retain stem-like features. Please see reply to reviewing editor comments above.

*2) The methods to generate "GBM stem cells" by in vitro introduction of multiple mutations into NSCs are less than ideal. These cells could lose homeostatic contact inhibition thus proliferate uncontrollably rather than bona fide tumor cells. Since the premise of the paper is based on these cells, all the beautiful data could be artificial rather than biological. Of course, data with human GBM cells greatly alleviate this concern. Still, it would be wise for authors to tone down the glioma claim with these in vitro transformed mouse NSCs.*

We have shown that GSC1 and 2 possess all characteristics of glioma stem-like cells, including self-renewal, differentiation into all three lineages and ability to form tumours with histopathological characteristics and transcriptional profiles of GBM when injected at clonal dilutions into recipient mice (Figure 1 and Figure 1—figure supplement 1). These experiments strongly indicate that the in vitro transformed NSC have GSC-like properties. Similar models have been previously developed by others and are accepted GBM models (Bachoo et al., Cancer Cell 2002, Blouw et al., Cancer Cell 2003). Nonetheless, as suggested by the reviewer we have toned down the claim that they are bona fide GSC cells throughout the manuscript.